# Sensitivity of the pseudo-global warming method under flood conditions: A case study from the Northeastern U.S.

Zeyu Xue[1,2], Paul Ullrich[1], and Lai-Yung Ruby Leung[2]

[1]Atmospheric Science Graduate Group, University of California, Davis, California, USA
[2]Pacific Northwest National Laboratory, Richland, Washington, USA

**Correspondence:** Paul Ullrich (paullrich@ucdavis.edu); Lai-Yung Ruby Leung (ruby.leung@pnnl.gov)

**Abstract.** Intensified extreme precipitation and corresponding floods are the most relevant consequences of climate change over the northeastern US (NEUS). To evaluate the impacts of climate change or certain climate perturbations on future extreme weather events which are dynamically similar to historic analogs, the pseudo-global warming (PGW) method has been frequently employed; however, this method lacks precise definition and guidelines, thus limiting its application. More specifically, three key questions related to the application of the PGW method remain unanswered: At what spatial scale should climate perturbations be applied? Among the different meteorological variables available, which ones should be perturbed? And will PGW projections vary significantly when different perturbations are applied? To address these questions, we examine the sensitivity and robustness of conclusions drawn from the PGW method over the NEUS by conducting multiple PGW experiments with varied perturbation spatial scales and choice of perturbed meteorological variables. The results show that the projections of precipitation and other essential variables at the regional mean scale are consistent across the PGW simulations, with a relative difference of much less than 10%; however, different perturbation modifications can cause significant displacements of the storm events being simulated. Several previously assumed advantages of modifying only the temperature at regional mean scale, such as the preservation of geostrophic balance, do not appear to hold. Also, for these experiments, we find the regional mean perturbation produces a positive precipitation bias because it ignores the land-ocean warming contrast, which is a robust regional response to global warming. Overall, PGW experiments with perturbations from temperature or the combination of temperature and wind at the gridpoint scale are both recommended, depending on the research questions. The first approach can isolate the spatially-dependent thermodynamic impact, and the latter incorporates both the thermodynamic and dynamic impacts.

## 1 Introduction

Historical observations and model projections highlight the significant risk that climate change poses for society (Pachauri et al., 2014). Among the many consequences of climate change, the intensification of extreme precipitation is considered to be one of the most impact-relevant (Pfahl et al., 2017). Within the United States, confidence is highest that the northeastern U.S. (NEUS) will experience the most significant intensification of extreme precipitation, with observations indicating that the most intense daily precipitation events in this region (those above the 99th percentile of daily precipitation) increased by more

than 70% from 1958 to 2012 (Melillo et al., 2014; Frumhoff et al., 2007; Kharin et al., 2007; Wuebbles et al., 2017; Pörtner et al., 2022). Accordingly, severe flood risk is also increasing, carrying with it increased risk of loss of life and infrastructural damage or failure (Hirabayashi et al., 2013; Frumhoff et al., 2007; Narayan et al., 2017). The NEUS is at especially high risk, considering it is the most populated and developed region in the U.S. (Hobbs, 2008; US Bureau of Economic Analysis, 2016). Therefore, robust and reliable projections of extreme precipitation and associated flooding are urgently needed in this region for adaptation planning.

Climate models are the most widely used tool for projecting climate change and its impacts (Kharin et al., 2007; Wagener et al., 2010; Frumhoff et al., 2007). Although significant progress has been made in improving models' physical consistency, persistent issues include large uncertainties and insufficient model resolution for representing extremes and regional impacts (Xu and Yang, 2012; Deser et al., 2020; Dai et al., 2020). These uncertainties mainly consist of scenario uncertainty, model uncertainty, and internal variability (Xie et al., 2015; Hawkins and Sutton, 2009; Deser et al., 2012). Among these, internal variability is generally deemed the most significant source of uncertainty in the near term and accounts for approximately half of the inter-model spread across North American precipitation projections in the next fifty years (Deser et al., 2020, 2014).

Climate model experiments with the Pseudo-Global Warming (PGW) method enable targeted exploration of regional impacts from future climate change while avoiding the large ensembles typically required to address internal variability (Schär et al., 1996; Xue and Ullrich, 2021b). Unlike traditional dynamical downscaling, which drives regional climate models (RCMs) using the outputs from global climate models (GCMs) to produce regional climate projections, the PGW method employs initial and boundary conditions from reanalysis data, perturbed using estimates of the mean changes from GCMs. The PGW approach ensures that the simulated weather essentially follows a similar time sequence and track as historical events, which can then be used to directly address questions about how climate change would affect a given weather event observed in the past if the same event and large-scale circulation pattern were to return in the future. The PGW framework allows for significant flexibility with respect to the choice of spatial scale and the choice of variables to be perturbed by the GCM climate change signals. Initially, the PGW method was employed to examine how atmospheric moisture and precipitation respond to a regionally uniform temperature increase (Schär et al., 1996; Frei et al., 1998).

In these first PGW studies, only a uniform temperature perturbation was added to the initial and boundary conditions derived from reanalysis because global warming represents the leading order behavior from climate change. Later studies extended this approach using a temperature perturbation that depended on both time and pressure (Hill and Lackmann, 2011; Yates et al., 2014; Mallard et al., 2013b, a; Ullrich et al., 2018; Xue and Ullrich, 2021b). By using a perturbation that was constant along pressure surfaces, it was anticipated that potential loss of geostrophic balance in the boundary conditions arising from inconsistency between the dynamical and thermodynamical fields could be avoided (Schär et al., 1996; Frei et al., 1998; Blumen, 1972). Additionally, a regionally-homogeneous perturbation applied only to the temperature field was assumed not to impact the dynamical fields, since geostrophic wind speeds are determined by the unmodified geopotential gradients. This approach was designed to avoid the well-documented uncertainties that persist in future projections of dynamical fields (Vecchi and Soden, 2007; Garner et al., 2009). Nonetheless, others have performed PGW simulations that modify both the thermodynamic and dynamical variables at each gridpoint to better reflect the dynamical influence of climate change (Kimura et al., 2007;

Kawase et al., 2009; Liu et al., 2017; Denamiel et al., 2020). Altering the dynamical fields (i.e., the wind speed and direction) has the potential to displace extreme weather events and so provide insights into how climate change would alter the tracks of these features. However, concerns have persisted that this approach could generate spurious gravity waves from the boundary of the simulation domain. In our simulations, gravity waves are more apparent in PGW simulations that include perturbations of the dynamical fields than in the historical runs. We hypothesize that the gravity waves originate from the inconsistencies and geostrophic inbalances in the dynamical and thermodynamic fields between the inner and outer domains. While these waves are filtered to some degree by numerical and physical viscosity in the regional climate model, care should be taken to ensure that this unphysical noise does not contaminate the simulation. Further study is needed to fully understand this issue.

The most significant advantage of the PGW method is that it captures the large-scale circulation of events that have been observed historically, and so avoids the uncertainty that a particular event in the traditionally downscaled simulations is a product of model biases in the GCM. These biases in the large-scale circulation forcing provided by GCMs are generally deemed to be one of the most significant sources of biases in dynamical downscaling (Wang et al., 2004; Sato et al., 2007). Furthermore, because PGW simulations are driven by historical reanalysis data plus the meteorological perturbations caused by climate change, they can directly answer the crucial question "what will historical extreme weather events look like under climate change?" This has led to the growing popularity of the PGW method since hazard management such as flood control often uses "model events" that are based on infamous historical disasters (Burns et al., 2007; Milly et al., 2008). The PGW method has a number of additional advantages, such as only requiring monthly mean GCM projections to provide climate perturbations, making it much easier to employ in conjunction with the large ensemble or multi-model GCMs.

Although the PGW method has grown in popularity in the past decade, there is still little guidance on best practices for the use of the PGW method. As noted already, the PGW method offers substantial flexibility in how it is employed: there is freedom to choose which boundary conditions to modify, whether or not the perturbations have space and time dependence, and how other inputs such as land use, greenhouse gas concentrations, or aerosols are modified. While these choices have led to substantial divergence in experimental design throughout the PGW literature, it is entirely possible that different choices may produce different conclusions. Altogether, these observations suggest the need for a sensitivity analysis of different PGW methods to examine the uncertainties and robustness of PGW simulations and to subsequently provide guidance on PGW experimental design. However, a single study cannot comprehensively evaluate all aspects of model sensitivity when employing PGW over multiple regions of different climatological character that considers domain size, choice of reanalysis data, modified variables, the temporal and spatial dependence of the climatological perturbations, regional model tuning parameters, and other such options. In this paper, we consider a narrower scope and focus our investigation on a single case study, with the expectation that this work could frame future investigations on this topic.

Given that storms and subsequent flooding are the most popular events in the PGW literature, and are known to be sensitive to both thermodynamic and dynamic changes (Frei et al., 1998; Kawase et al., 2009; Knutson et al., 2013; Mallard et al., 2013b; Yates et al., 2014; Prein et al., 2017; Rasmussen et al., 2020; Dougherty and Rasmussen, 2020, 2021), they provide a suitable context for investigating sensitivities in the PGW method. In this paper, we perform an ensemble of PGW simulations that involve three major flood events over NEUS, each the product of different meteorological drivers, including the October 2005

flood (October 7th to 17th), the New England flood of May 2006 (May 12th to May 20th) and the 2006 Mid-Atlantic United States flood (June 23rd to July 5th). In the following analysis and the corresponding figures, "2055 October", "2056 May" and "2056 June" refer to the flood periods of October 7th to 17th 2005, May 12th to May 20th, 2006 and June 23rd to July 5th, 2006, respectively, in the PGW experiments for the perturbed climate of 2055/2056. The period mean refers to the average of these three periods. Our simulation ensemble includes variations of the modified variables (temperature, wind, geopotential height, and sea surface pressure) and the method with which the modification is applied (application of the perturbation as a regional mean or for each gridpoint). We focus primarily on comparing those simulations where temperature is modified at the regional mean scale or at each gridpoint, and comparing simulations with perturbations applied to different meteorological variables at each gridpoint. This paper aims to answer the following major questions: First, are PGW simulations sensitive to the spatial scale of climate perturbations? Second, besides temperature, which meteorological variables should be modified in PGW experiments? Lastly, we summarize our results to provide some guidance for the design of PGW simulations.

## 2 Data and methods

### 2.1 Experiments design

Our simulations use the Weather Research and Forecasting (WRF) model 3.9 (Skamarock et al., 2008; Powers et al., 2017) with hybrid mass-based coordinate to simulate the 16-month period between April 2005 and July 2006. This period is chosen as it includes the three major flood events in Table 1. Historical and future simulations are driven by reanalysis data without and with perturbations to meteorological variables obtained from GCMs. WRF has been used extensively for regional climate modeling, demonstrating reasonable fidelity in reproducing regional climatology (Knutson et al., 2013; Heikkilä et al., 2011; Rasmussen et al., 2011; Liu et al., 2017; Beck et al., 2019; Dai et al., 2020). Our simulations use the parameterization set employed in our previous PGW studies (Table S1) (Ullrich et al., 2018; Xue and Ullrich, 2021b), as this choice has been demonstrated to be both robust and produce good performance over the NEUS. Additionally, all simulations use the Community Land Model. An investigation of the sensitivity of the PGW simulations to the parameterization set and land model is beyond the scope of the present study, as we expect this choice to be orthogonal to our conclusions.

All PGW experiments cover two periods: the historical flood period (2005 April to 2006 July) and the mid-21$^{st}$ century "returned" flood period (2055 April to 2056 July). Following (Jerez et al., 2020), the first six months of the simulation serve as the spin-up period to ensure consistency between the meteorology and land surface states. Two nested domains are employed that cover the NEUS, with resolutions of 30 and 10 km respectively, as shown in Figure 1. In this study, without notation, our analysis is based on the inner domain, which covers most of the NEUS at the finer resolution. Our analysis of sea level pressure includes the whole domain because of its connection with the broader atmospheric circulation. In our WRF simulations, spectral nudging is employed using the default relaxation timescale (the guv, gt, gq and gph are equal to 0.0003; the xwavenum and ywavenum are equal to 3). 30-arc second resolution United States Geological Survey-based geography data provides geographic data such as land use, elevation, green fraction, and leaf area index data.

**Table 1.** Characteristics of the three flood periods studied in this paper.

| Flood Period | Duration | Meteorological Cause | Region |
|---|---|---|---|
| Northeast U.S. flood of October 2005 (2005 Oct Flood) | October 7th to 17th, 2005 | A larger extratropical storm in the eastern Gulf of Mexico absorbed a part of Tropical Storm Tammy and connected with a cold front system over the Mid-Atlantic region then a low-pressure center staying over the Long Island dragged a large amount of moisture from the western Caribbean Sea and brought heavy rainfall to the NEUS (Stewart, 2006; Beven, 2006; Oravec, 2006; Station, 2005; Stuart and Grumm, 2009). | The interior of New England, as well as over parts of New Jersey and New York. |
| New England Flood of May 2006 (2006 May Flood) | May 12th to May 20th, 2006 | An unusually strong low-pressure system that stalled over the central United States pulled large amounts of moisture from the Atlantic Ocean to the NEUS (Center, 2006c, d; Stuart and Grumm, 2009; Agel et al., 2015). | New England, especially in New Hampshire and Massachusetts. |
| 2006 Mid-Atlantic United States flood (2006 June Flood) | June 23rd to July 5th, 2006 | The tropical low over the North Carolina coast brought constant tropical moisture to the Mid-Atlantic region which is blocked by the stalling of the jet stream over the west of Appalachian Mountains and the Bermuda High over the Atlantic Ocean thus formed large amounts of rainfall (Center, 2006a, b; Stuart and Grumm, 2009). | The Mid-Atlantic region. |

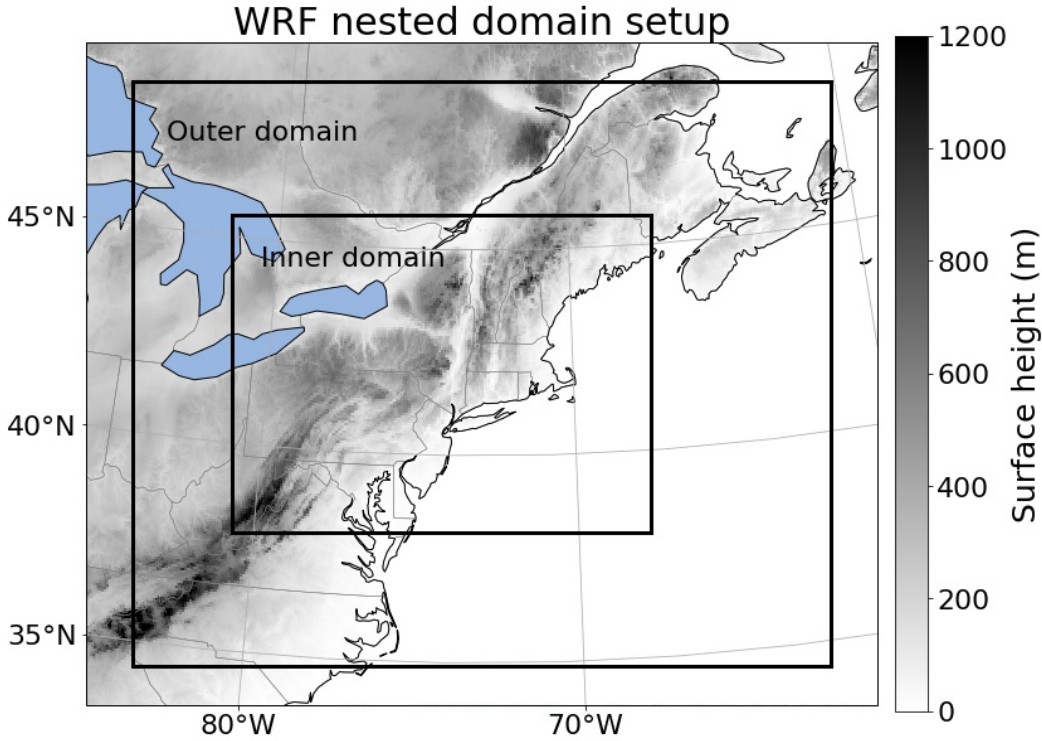

**Figure 1.** The WRF domain for all simulations in this study. Shading indicates the surface elevation. Grid spacing in the outer (inner) domain is 30 km (10 km).

## 2.2 Methodology and modified forcings

The 6-hourly ECMWF Reanalysis 5th Generation (ERA5) data is employed to provide initial and boundary conditions. ERA5 is a next-generation reanalysis product that replaces the ERA-Interim reanalysis (European Centre for Medium-Range Weather
Forecasts, 2020; Hersbach et al., 2020), incorporating improved data assimilation, core dynamics, model physics, temporal and spatial resolution. ERA5 has been shown to represent low-frequency variability well, and so is expected to capture historical meteorological conditions with high fidelity (Hersbach et al., 2020; Tarek et al., 2020; Dullaart et al., 2020).

For our future PGW simulations, the initial and boundary conditions derived from ERA5 are adjusted by adding the monthly mean and ensemble mean climate perturbations from the Community Earth System Model (CESM1) Large Ensemble (LE)
dataset (National Center for Atmospheric Research, 2020) under Representative Concentration Pathway (RCP) 8.5. The climate perturbations are calculated by taking the difference between the meteorological fields from 2030-2059 and 1980-2009. These long periods are used to reduce noise from internal variability. In all cases, perturbations are computed for each calendar month (i.e., each of January, February, etc. has its own climatological perturbation), and linearly interpolated in time (between the middle of two consecutive months). CESM1 is employed here since it is a high-quality model with demonstrable
performance over the NEUS (Kay et al., 2015; Swann et al., 2016; Sillmann et al., 2013; Karmalkar et al., 2019; Xue and

Ullrich, 2021a). Further, the CESM1 initial condition ensemble contains forty ensemble members that reasonably capture the internal variability, and all necessary meteorological variables for this study are available from the model output. Moreover, since the parameterization set and land model in our WRF simulations are also used in CESM, we can maintain some degree of consistency between the forcing data and the regional climate simulation.

## 2.3 Ensemble design

Following (Schär et al., 1996) and the numerous PGW studies since (Kawase et al., 2009; Knutson et al., 2013; Mallard et al., 2013b; Yates et al., 2014; Xue and Ullrich, 2021b; Ullrich et al., 2018), we conduct and compare five PGW experiments that vary the modified meteorological fields at the gridpoint level, and one additional experiment that applies the thermodynamic modification at the regional mean scale. All experiments investigated are described in Table 2. 3D variables modified in these experiments are air temperature (T), zonal and meridional wind (UA and VA), and geopotential height (ZG). 2D variables modified in these experiments are sea-level pressure (SLP) and sea surface temperature (SST). In all PGW simulations, relative humidity is assumed constant, and so specific humidity is updated as a function of the modified temperature. Constant relative humidity is a common assumption used in PGW experiments since it is largely unaffected by climate change in moist regions, particularly in the lower troposphere. Greenhouse gas concentrations are also updated in WRF, in accordance with those prescribed by the RCP8.5 emission scenario.

## 3 Result

### 3.1 Model validation

Before examining the sensitivity of different PGW simulations to experimental design, we first validate the ability of our historical WRF simulation to simulate the observed surface temperature and precipitation during the events of interest. WRF is the most widely-used regional climate model in the PGW literature and has consistently demonstrated good performance for simulating regional climate when using appropriate parameterizations (Kharin et al., 2007; Denamiel et al., 2020; Lackmann, 2015), and so we keep our validation brief. For comparison, we use the Climate Prediction Center unified gauge-based analysis precipitation data (CPC) (NOAA Physical Sciences Laboratory, 2020), the Integrated Multi-satellite Retrievals for GPM (IMERG) (Huffman et al., 2015; The National Aeronautics and Space Administration, 2021), and the driving reanalysis (ERA5) (European Centre for Medium-Range Weather Forecasts, 2020). Fig. 2 shows that the simulated temperature at 2 meter is similar to its driving data (ERA5) except over the lake regions (note that the CPC daily 2-meter temperature is derived from the mean of daily minimum and maximum). In Fig. 3, it's clear that WRF's simulated precipitation is similar in magnitude and structure to the CPC, IMERG and ERA5 data, although displacements exist. The finer grid spacing of the WRF simulation further enables us to better capture the locations of high precipitation intensity. Based on this comparison, it's clear that both the surface temperature and precipitation match well with observations during the three flood periods. Although some biases exist,

**Table 2.** Ensemble Design

| Simulation | Modified variables | Perturbation scale |
|---|---|---|
| PGW_T_ regional | Air temperature at each pressure level | Regional mean |
| | Sea surface temperature | |
| PGW_T_gp | Air temperature at each pressure level | Gridpoint |
| | Sea surface temperature | |
| PGW_T_WIND_gp | Air temperature at each pressure level | Gridpoint |
| | Wind at each pressure level | |
| | Sea surface temperature | |
| PGW_T_ZG_gp | Air temperature at each pressure level | Gridpoint |
| | Geopotential height at each pressure level | |
| | Sea surface temperature | |
| PGW_T_SLP_gp | Air temperature at each pressure level | Gridpoint |
| | Sea level pressure | |
| | Sea surface temperature | |
| PGW_T_WIND_ZG_SLP_gp | Air temperature at each pressure level | Gridpoint |
| | Wind at each pressure level | |
| | Sea level pressure | |
| | Sea surface temperature | |

as the aim of this study is to examine the response of the PGW simulations to different perturbation modification methods, the simulation accuracy itself is not the primary concern in this study.

## 3.2 PGW simulations with regional mean vs. gridpoint temperature perturbations

Given the direct impact on near-surface temperatures and the indirect impact on precipitable water content, modifying temper-
175 ature is expected to have the most significant effect in PGW experiments. As mentioned earlier, while some past studies apply a regional mean perturbation to temperature (1D space + 1D time), others apply temperature perturbations at every grid point (3D space + 1D time). In this section, we investigate the difference between these two methods.

Figure 4 shows the flood period mean temperatures from the simulation with regional mean perturbations (PGW_T_regional) and the temperature difference between the simulation with gridpoint perturbations (PGW_T_gp) and (PGW_T_regional). As
expected, the regional mean simulation exhibits more uniform warming, especially when looking over the whole simulation period (Fig. S1). However, this outcome is somewhat inconsistent with the GCM projections and PGW_T_gp, which generally indicate enhanced warming over land and suppressed warming over the ocean as a consequence of differences in heat capacity and water availability and the redistribution of temperature during storm events (the left column in Fig. 4). Precipitation amounts are subsequently affected by this difference: As the ocean is the primary source of moisture of storm events (Fig. S14),

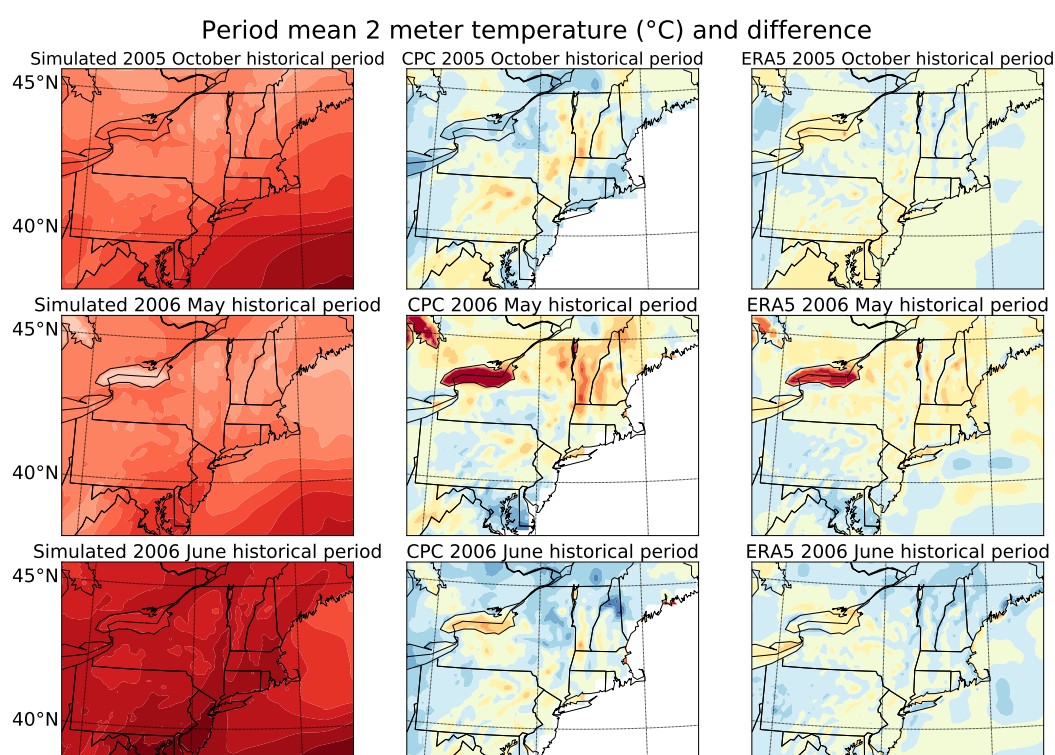

**Figure 2.** (Left) Period mean 2-meter temperature (°C) over the inner domain from the historical simulation. (Right) Differences between the period mean 2-meter temperature (°C) from the CPC observational data, IMERG and ERA5 reanalysis data and the historical simulation shown on the left column.

and most precipitation occurs over the coast (Fig. 5), the simulation with regional mean temperature perturbations produces more precipitation in the future than the gridpoint perturbation experiment (i.e., regional period mean precipitation increases from 12.69 mm d$^{-1}$ to 13.86 mm d$^{-1}$ in the 2005 Oct Flood, 7.19 to 7.83 mm d$^{-1}$ in the 2006 May Flood, and 7.43 to 7.88 mm d$^{-1}$ in the 2006 June Flood, as in Fig. 6). Notably, the first flood event (October 2055) exhibits a greater difference between regional and gridpoint perturbation experiments (9.22 %) compared with the other two events (3.38 % and 0.61 %). Nonetheless, the relative displacement of the storm pattern is more apparent than the regional mean precipitation differences. As the scaling of extreme precipitation with temperature can be decomposed into components associated with the vertical pressure velocity and the vertical derivative of the saturation specific humidity (Pfahl et al., 2017), the greater impact of experimental design on the first storm event appears to be a result of two factors: first, the strong frontal system associated with this event has a more intense uplift (Fig. S10); and second, the strong on-shore flow (Fig. S9) associated with this event greatly amplifies the on-shore water vapor transport by the storm. The vertical pressure velocity is not significantly modified in each PGW simulation, suggesting

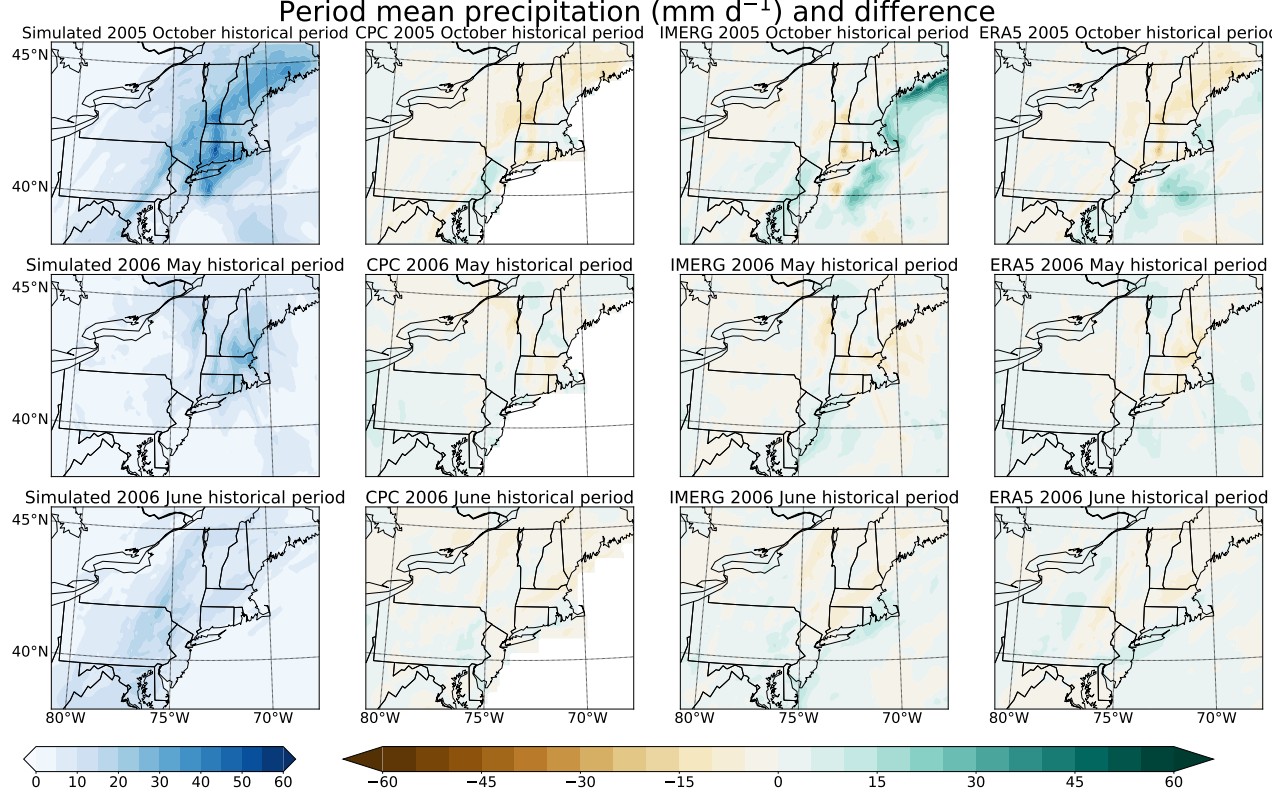

**Figure 3.** (Left) Period mean precipitation (mm d$^{-1}$) over the inner domain from the historical simulation. (Right) Differences between the period mean precipitation (mm d$^{-1}$) from the CPC observational data, IMERG and ERA5 reanalysis data, and the historical simulation shown on the left column. All datasets have been interpolated to the resolution of our inner domain (10km).

that the increase in precipitation is primarily thermodynamic (Norris et al., 2019). Additionally, as the frontal system's intensity is highly dependent on the horizontal temperature gradient (Sawyer, 1956; Bosart, 1975; Reeder et al., 2021), it's intuitive that the first flood event is more sensitive to the different spatial scales of temperature perturbations applied to the PGW runs.

If we look at the entire simulated period, the PGW_T_regional simulation has a far larger regional mean precipitation increase compared with PGW_T_gp over the sea (with 12.12% relative change) than over the land (with 2.49% relative change). This result is consistent with the larger warming over the ocean than land in PGW_T_regional relative to PGW_T_gp, so with relative humidity held constant, there is more precipitable water and vapor transport in the regional mean simulation over the ocean, but less over the land (Fig. S2). This observation also explains why the third flood event has the smallest precipitation increase since it features more inland precipitation than the other two (Fig. 5). The period maximum regional mean precipitation also reflects these observations (Fig. 6).

### 3.3 Gravity wave noise

As mentioned earlier, one concern with using gridpoint level perturbations is the possible generation of gravity wave noise due to geostrophic adjustment. However, high-frequency wave-like noise (i.e., transient inertia-gravity waves) is apparent in the meteorological field examined (such as the sea level pressure) in both PGW_T_regional and PGW_T_gp at approximately the same magnitude, if the fields are examined on hourly scales (animations of this noise are available at Xue and Ullrich (2022)). This contradicts a common assumption in previous studies that regionally-uniform or latitudinally-uniform temperature perturbations inherently avoid geostrophic imbalance and the resulting adjustments (Schär et al., 1996; Frei et al., 1998; Hill and Lackmann, 2011; Yates et al., 2014; Mallard et al., 2013b; Ullrich et al., 2018). However, none of the PGW studies cited identified that even applying a uniform temperature perturbation can induce meteorological noise in PGW simulations. This is likely because the noise is not apparent upon taking the period mean (Fig. S3), which is generally the focus of previous PGW studies (Lackmann, 2013; Mahoney et al., 2018).

In WRF, boundary conditions are specified based on the input forcing (here the historical reanalysis data with climate perturbations) and the numerical solutions are nudged towards the imposed boundary conditions within the buffer zone of the outer-most domain (Skamarock et al., 2008). Within the inner domain, except for the most peripheral grid points, meteorological fields are much less constrained by the boundary conditions and can significantly depart from the forcing data (Skamarock et al., 2008). This can induce inconsistencies among the meteorological fields in the outer and inner domains and explain why the wave-like noise also appears in PGW_T_regional (refer to the animations at Xue and Ullrich (2022)), even though the geostrophic balance holds in the initial and boundary conditions. From the animations of SLP during the October flood event (Xue and Ullrich, 2022), we can infer that the gravity wave is not solely excited by the storms, as the magnitude of the gravity wave is negligible in the historical run compared to the PGW run even though the storm is also present in the historical run. Furthermore, gravity waves are amplified in the PGW run more than expected from the difference in the precipitation between the PGW and historical runs. Similar results can be observed in the magnitude spectrum of SLP (Fig. S15). Storms also play an essential role in magnifying the gravity waves through significant advection of energy and momentum, as it is apparent that gravity waves are much stronger during storms than during periods with nearly no precipitation (refer to the animations at Xue and Ullrich (2022)). We conclude that the amplified gravity waves in the PGW run during storm events reflect the interactive effect of inconsistency between the meteorological fields in the outer and inner domains and the excitation of gravity waves by the storms. Although gravity waves in the PGW simulations are inspected through hourly animations and magnitude spectrum, a comprehensive and exhaustive analysis of gravity waves could be its own standalone study

Moreover, while it is normally assumed that dynamical fields are unchanged under a uniform temperature perturbation (Schär et al., 1996; Frei et al., 1998; Hill and Lackmann, 2011; Yates et al., 2014; Mallard et al., 2013b; Ullrich et al., 2018), we notice that both the wind and sea level pressure of PGW_T_regional deviate from the historical run (Fig. S11 to S13) and the differences are even comparable to those in PGW_T_gp because of the redistribution of energy and momentum during the weather events. For example, temperature advection over the ocean during the storm will bring more energy to the coastal region and so can intensify local convection processes, reduce the sea level pressure and alter the wind fields. Since PGW_T_regional

may overestimate precipitation because it does not capture the robust land-ocean warming contrast, and it does not obviously reduce gravity wave noise, we do not recommend this experimental configuration.

## 3.4 Sensitivity of PGW simulations to choice of perturbed meteorological fields

We now turn our attention to comparing the simulation responses when different sets of climatological fields are modified at the gridpoint level. In this case, the baseline simulation only applies temperature perturbations at each gridpoint and the effect of modifying other meteorological fields is then ascertained by comparing it with the baseline. As we can see in Fig. 6, perturbations to the meteorological fields produce a spread in regional precipitation rates and totals; however, at the regional mean scale, this spread is fairly small across simulations, with different choices producing precipitation differences within 10% of the baseline. However, we do observe that the October flood event is more sensitive to the experimental setup than the other two events. The reasons for this difference will be discussed in the following sections.

### 3.4.1 Sensitivity of PGW simulations to inclusion of wind perturbations

When wind perturbations are included, all PGW simulations generally simulate more precipitation over land, especially along the coast of the inner domain. The coastal region also experiences the most rainfall in each historical period (Fig. 6 and 7) because precipitation in the coastal region is largely driven by the transport of precipitable water from the ocean to land (Fig. S2). As the land and sea areas are found in the northwestern and southeastern portions of our domain, respectively, both positive meridional wind and negative zonal wind perturbations over the sea increase the advection of precipitable moisture into the coastal region of the inner domain, in turn enhancing precipitation. For example, during the 2055 October flood period over the inner domain, the zonal wind is modified with a strong negative perturbation (with a regional mean of -0.15 m s$^{-1}$), much larger than the positive meridional wind perturbation (0.014 m s$^{-1}$), corresponding to enhanced onshore flow. This leads to more vapor advection and enhanced precipitation over the coastal region of the inner domain, with a 0.71 mm d$^{-1}$ regional mean land precipitation increase (as shown by the difference between the purple cross and green triangle in Fig. 6). Additionally, during this event, compared with PGW_T_gp, the addition of the wind perturbation reduces precipitation over the sea, as the wind anomaly carries moisture onto land. During the other two flood periods, both the meridional and zonal wind perturbations are much smaller than 0.1 m s$^{-1}$, so the subsequent land precipitation increase is also much smaller (a regional mean increase of 0.16 and 0.19 mm d$^{-1}$). The magnitude of these wind perturbations explains why, under this PGW experiment that includes wind perturbations, the first flood event shows a stronger future response. Considering the whole simulation period, there are slightly positive meridional wind perturbations (with a regional mean of 0.054 m s$^{-1}$ and 0.016 m s$^{-1}$ over the sea and land) and negative zonal wind perturbations (with regional mean of -0.039 m s$^{-1}$ and -0.023 m s$^{-1}$ over the sea and land), which results in a slight precipitation increase (with regional mean of 0.13 mm d$^{-1}$ over both the sea and land, referring to the difference between the purple cross and green triangle in Fig. 6). Since the wind perturbation conveys useful information pertaining to climate change's impact on atmospheric dynamics, we acknowledge it is useful to include in general PGW studies. However, as mentioned earlier, significant uncertainties persist regarding the future dynamical change, and so care should be taken when wind perturbation is included.

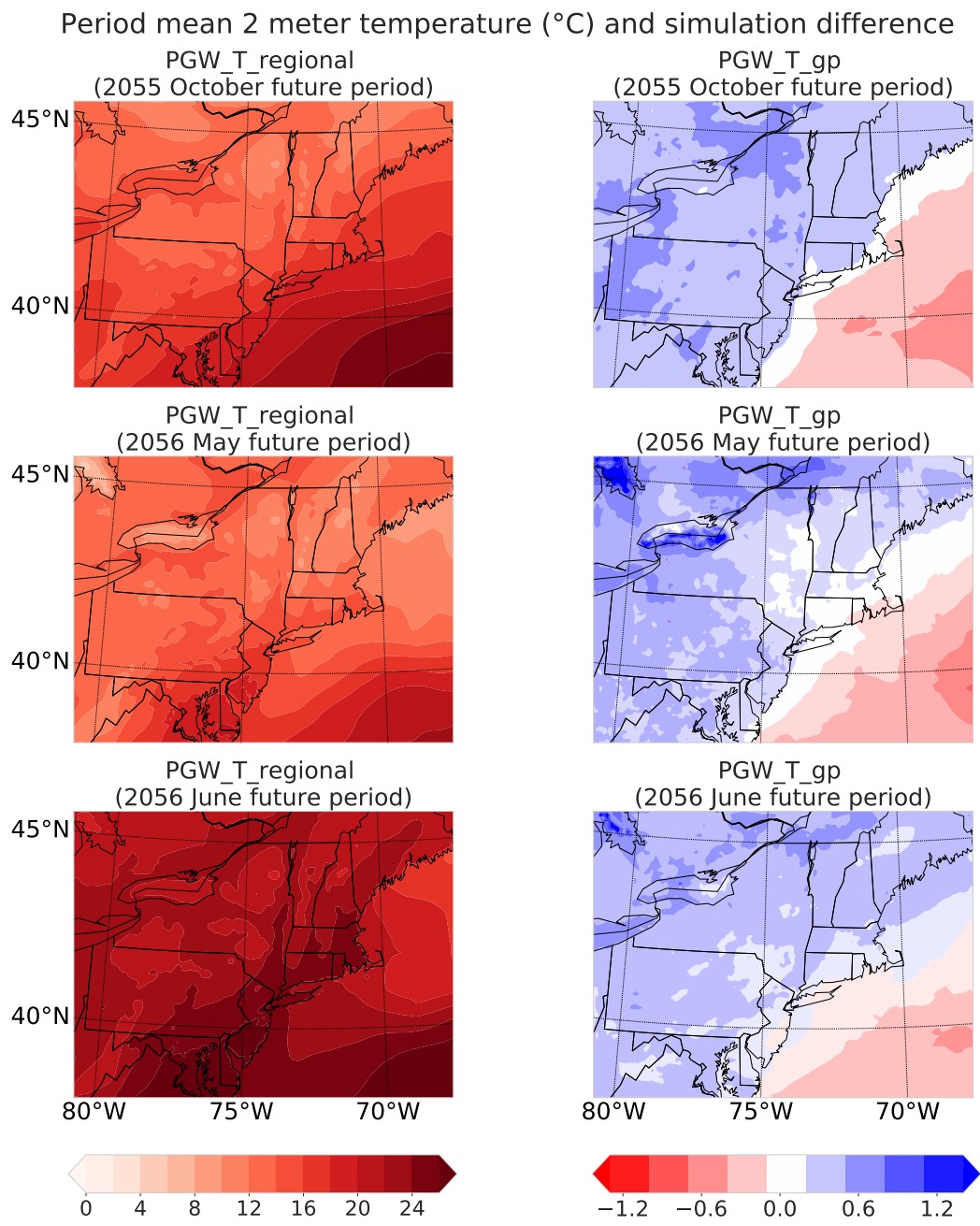

**Figure 4.** (Left) Period mean 2-meter temperature (°C) over the inner domain from the simulation with temperature perturbation at regional mean scale. (Right) Differences between the period mean 2-meter temperature (°C) from the simulation with gridpoint perturbation and the simulation with regional mean perturbation shown on the left column.

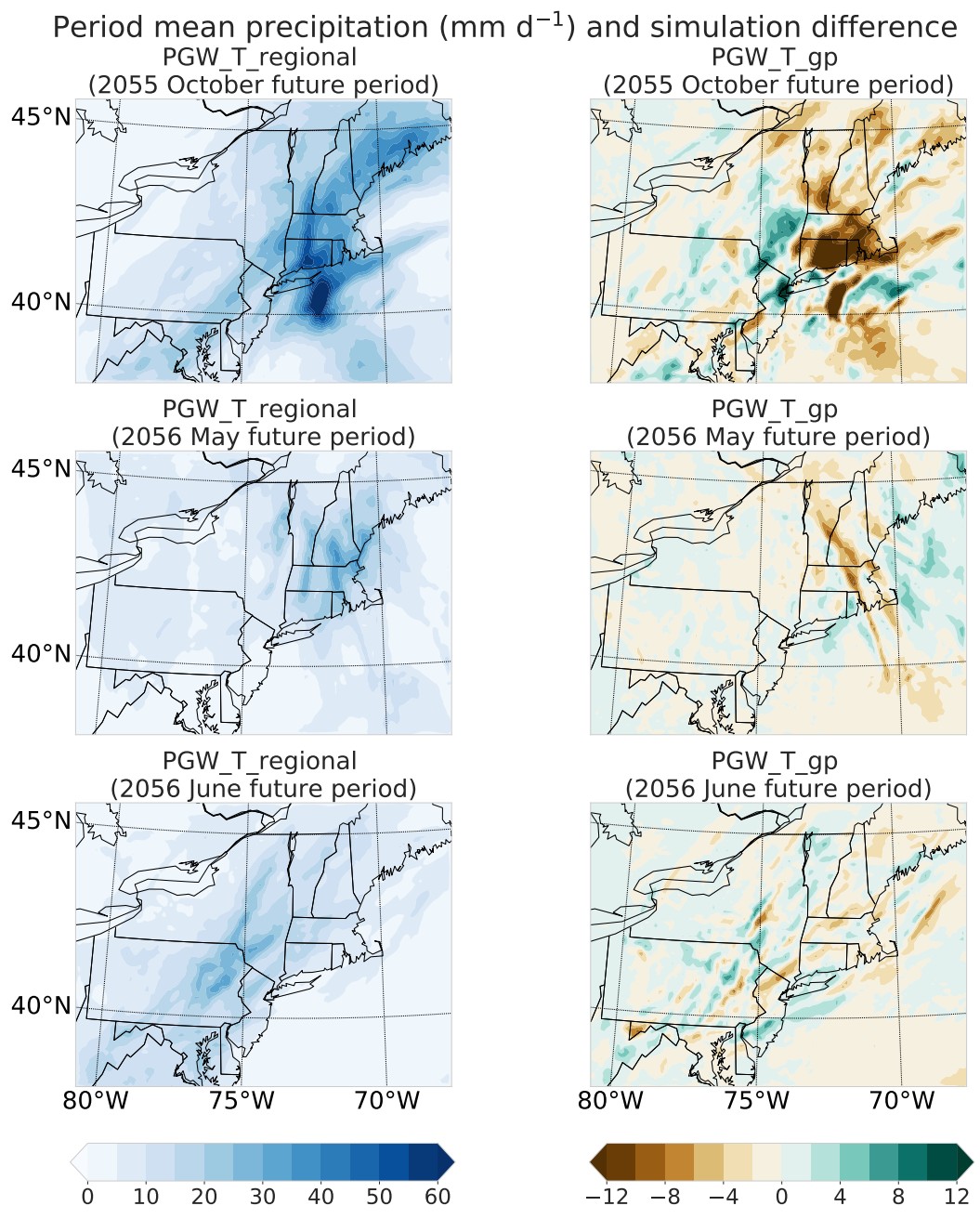

**Figure 5.** (Left) Period mean precipitation (mm d$^{-1}$) over the inner domain from the simulation with temperature perturbation at regional mean scale. (Right) Differences between the period mean precipitation from the simulation with gridpoint perturbation and the simulation with regional mean perturbation shown on the left column.

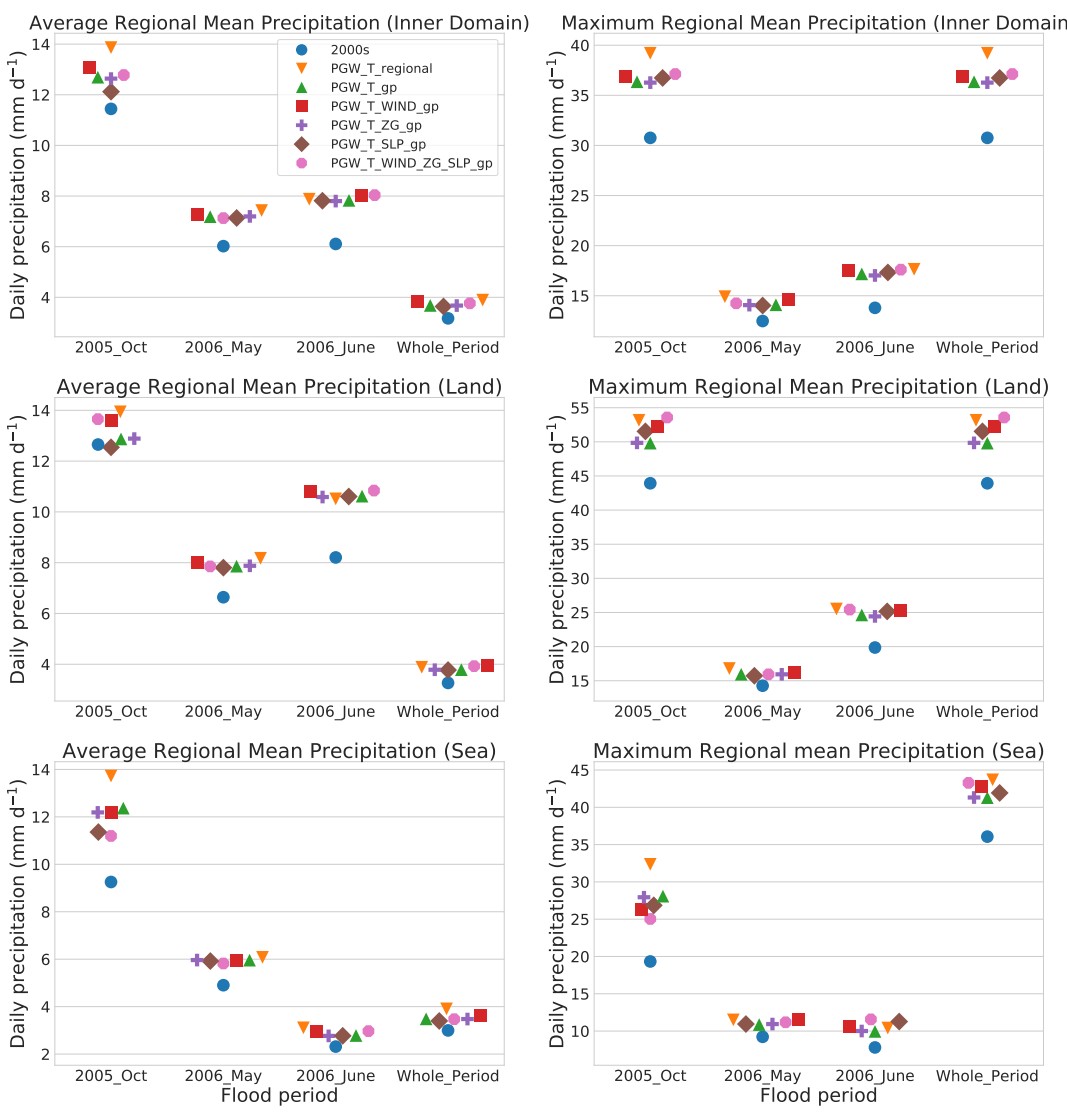

**Figure 6.** Period mean precipitation (mm d$^{-1}$) averaged over the inner domain (top) and the land area (middle) and sea area (bottom) within the inner domain from all PGW simulations with different perturbation modification methods.

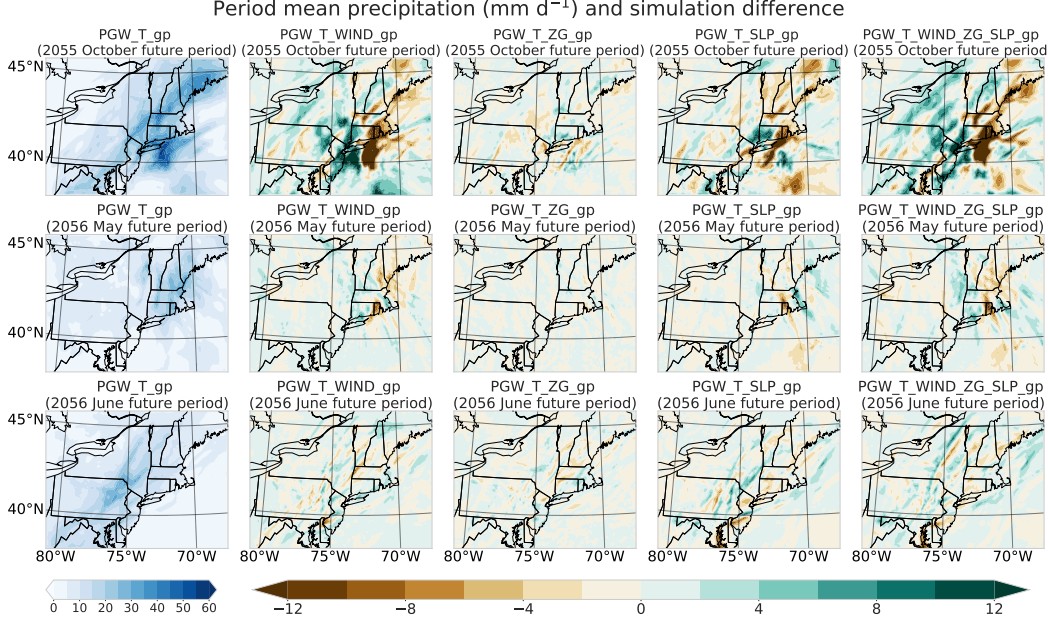

**Figure 7.** Period mean precipitation (mm d$^{-1}$) over the inner domain from the simulation with only temperature perturbation at each grid-point, and differences between simulations with additional dynamical perturbations of wind, geopotential height, sea level pressure and the combination of them at each gridpoint and the simulation with temperature perturbation at each gridpoint only.

### 3.4.2 Sensitivity of PGW simulations to inclusion of geopotential height perturbations

Among all the meteorological fields, modifying geopotential height (ZG) has the most insignificant impact on simulated precip-
275 itation: overall, the relative change in period regional mean precipitation is less than 1% (Fig. 6). This result is not unexpected, since the WRF Pre-Processing System (WPS) automatically applies vertical hydrostatic adjustments to produce a new geopotential height field that accords with the modified temperature fields via the hypsometric equation (Skamarock et al., 2008). Indeed, this is confirmed in Fig. 8, as we can see that even at 300 hPa, the simulation with geopotential height perturbations applied produces a similar geopotential height output to our baseline simulation. Even when examining individual gridpoints,
the simulation with modified ZG is largely indistinguishable from the baseline simulation (Fig. 7). Therefore, we do not recommend modification of the geopotential field in PGW studies, as it is unnecessarily redundant with temperature perturbation.

### 3.4.3 Sensitivity of PGW simulations to inclusion of sea level pressure perturbations

From Fig. 7, the simulation with modified sea level pressure consistently produces less precipitation over both the land and sea area of the inner domain during all three flood periods. This reduction in precipitation arises from a largely positive sea
level pressure perturbation over the whole sea area, which is the main source of moisture. From Fig. 9, in the simulations with modified SLP, SLP perturbations are generally positive over the coastal and ocean areas and negative over the inland

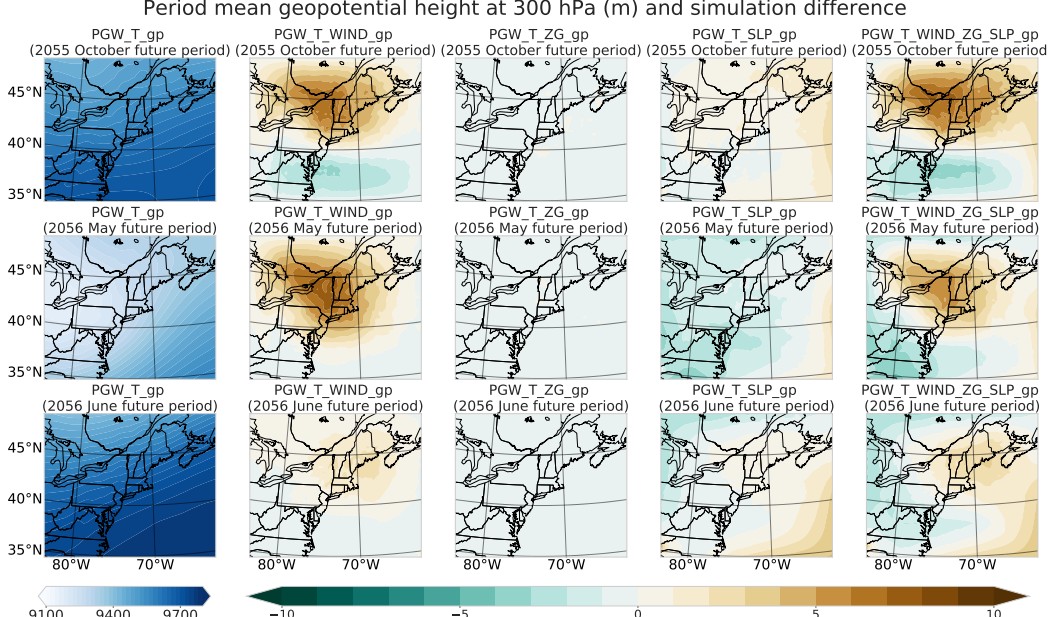

**Figure 8.** Period mean geopotential height (m) at 300 hPa over the whole domain from the simulation with temperature perturbation at each gridpoint, and the differences between simulations with additional dynamical perturbations of wind, geopotential height, sea level pressure and the combination of them at each gridpoint and the simulation with only temperature perturbation.

area. The magnitude of the positive SLP difference is larger during the first flood event (regional mean of 0.22 hPa) than the other two events (regional means of 0.027 hPa and 0.16 hPa, respectively). Although the third flood event has a comparable regional-mean SLP perturbation to the first flood event over the sea, its SLP difference is negative over the inland regions with
the highest precipitation, while the other two flood events exhibit positive differences. This results in the third flood event of the simulation with modified SLP having the smallest precipitation decrease over the inner domain (with the regional mean of -0.014 mm d$^{-1}$ compared to the other two events' -0.57 and -0.051 mm d$^{-1}$) (Fig. 7 and 6).

The most direct consequence of applying the SLP perturbation is a corresponding adjustment to regional near-surface temperature fields. This change emerges since WRF uses a hybrid vertical coordinate which is of roughly constant layer thickness
at ground level, and so near-surface changes to SLP are incorporated at constant volume. By the hypsometric equation, any increase (decrease) in SLP must be compensated for by a decrease (increase) in average layer temperature. Indeed, from Fig. 10, we can see that the simulation with modified SLP has significantly colder 2-meter temperatures in all three flood periods; however, the third flood period is slightly different from the other two periods in that it does show some temperature enhancement throughout the NEUS. As the relative humidity is fixed among all simulations, colder temperature fields directly induce
a reduction in precipitable water (Fig. 11). Overall, the regional period mean precipitable water decreases by 0.050, 0.083, and 0.027 mm in the three flood periods, respectively. The reduction in precipitable water over the sea is stronger than over the land except for the 2056 May flood period, probably because the SLP increase over the sea is smaller during this event (Fig.

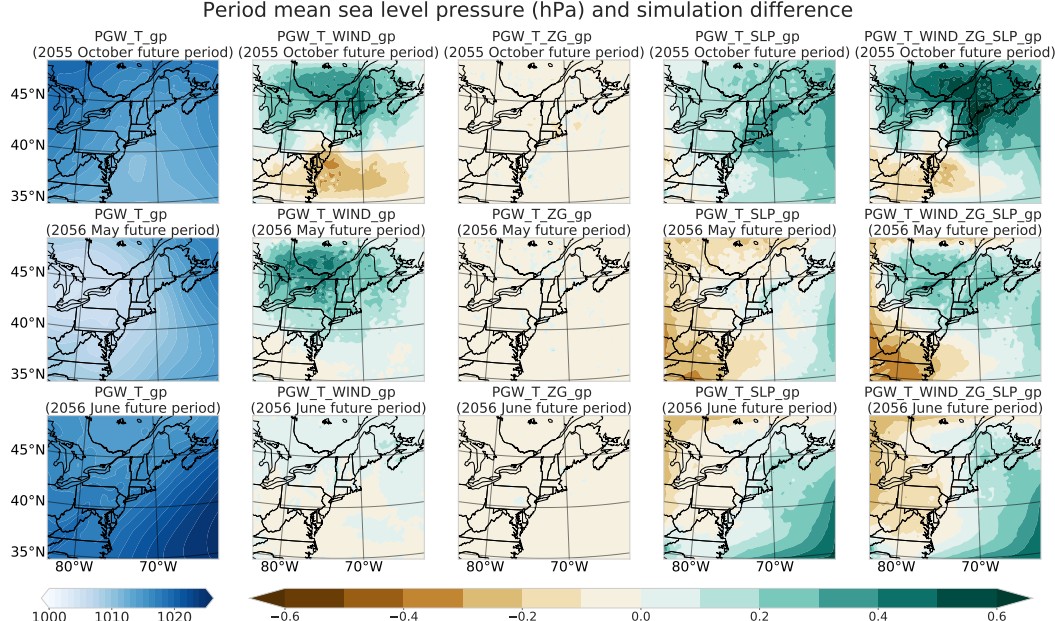

**Figure 9.** Period mean sea level pressure (hPa) over the whole domain from the simulation with temperature perturbation at each gridpoint, and differences between the simulations with additional perturbations of wind, geopotential height, sea level pressure and their combination at each gridpoint and the simulation with temperature perturbation only.

9). Another consequence of a positive sea level pressure perturbation is suppressed convection during all three flood periods – and indeed we see the relative decrease of the regional period mean convective available potential energy (CAPE) is 8.62%,

1.88%, and 5.34% over the ocean region (Fig. 12). In turn, this reduces convective precipitation over the inner domain (with regional mean of -0.39, -0.19 and -0.04 mm d$^{-1}$, where again the third flood event has the smallest decrease). Moreover, the modified SLP field will impact wind fields (Fig. S6 and S7) and produce more obvious wave-like noises (Xue and Ullrich, 2022). Altogether, the simulation with modified SLP produces less precipitation overall.

It's clear that the inclusion of the SLP perturbation significantly weakens the warming signals from LE CESM1 (Fig. 13). For

example, during the 2056 flood period, the regional mean temperature increase is 1.82 °C at 39th parallel north in PGW_T_gp; however, it drops to 1.00°C in the simulation with SLP perturbations (PGW_T_gp), which is less than half of the warming signal (2.11°C) at the same latitude provided by LE CESM1 (Fig. 13). Consistent with our explanation above, the underestimation of the warming signal is constrained to the lowest model levels (Fig. 14). While simulations not including the SLP perturbation are largely consistent in the near-surface with the LE CESM1 average, the inclusion of the SLP perturbation produces a strong

divergence between WRF and the driving data. Because of this anomalous behavior, we recommend not including the SLP perturbations in PGW experiments.

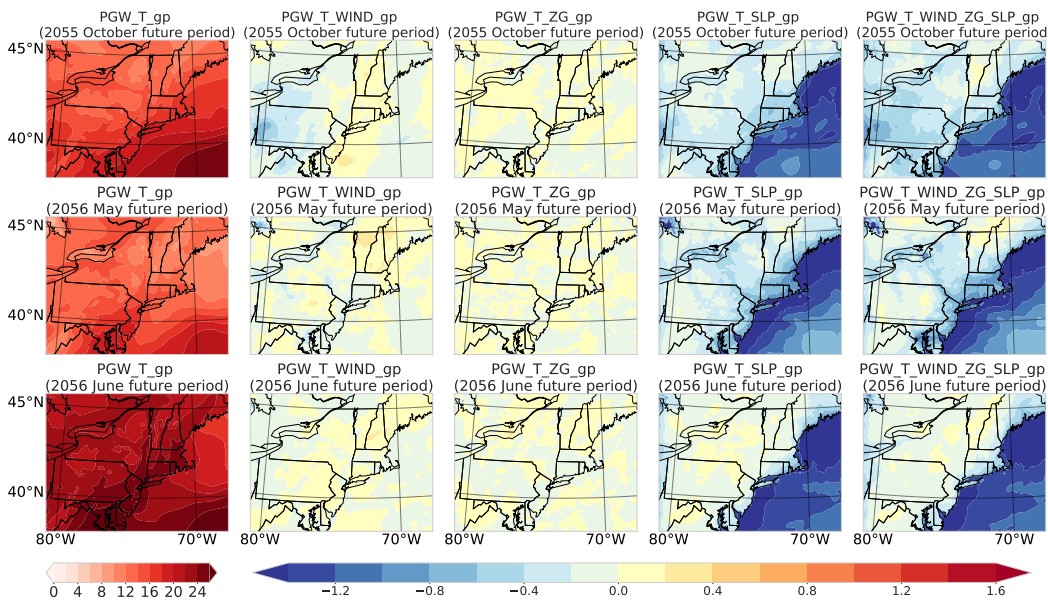

**Figure 10.** Period mean temperature (°C) at 2 meter over the inner domain from the simulations with temperature perturbation at each gridpoint, and the differences between the simulations with additional perturbations of wind, geopotential height, sea level pressure and their combination at each gridpoint and the simulation with temperature perturbation only.

## 4    Discussion

To better quantify the differences among PGW simulations, we employed the Kolmogorov-Smirnov (K-S) Test (Massey Jr, 1951), which is used to determine if two datasets have the same distribution, and the Student's t-test (Student, 1908), which is used to examine if two datasets have the same mean value, to compare all PGW simulations during the whole simulation period. We choose the PGW_T_gp as the baseline and compared it with other PGW simulations, focusing on the daily regional mean precipitation and 2-meter temperature. Results show that the p-values from both the K-S Test and t-test are much larger than 0.05 in all cases, which indicates that the null hypothesis that the daily regional mean precipitation and 2-meter temperature of PGW_T_gp and other PGW simulations have the same distribution and mean values cannot be rejected at the 95% confidence level. Further, we conducted similar statistical tests on the daily precipitation and 2-meter temperature at each gridpoint. The results show that, except for PGW_T_ZG_gp, the p-values of both the K-S Test and t-test between all other PGW simulations and PGW_T_gp are nearly zero (much less than 0.05), indicating evidence against the null hypothesis that the daily precipitation and 2-meter temperature at each gridpoint of PGW_T_gp and other PGW simulations (except PGW_T_ZG_gp) have the same distribution and mean values at the 95% confidence level. However, the p-values of these two tests on daily precipitation and 2-meter temperature at each gridpoint between PGW_T_gp and PGW_T_ZG_gp are still much larger than 0.05 (0.9997 and 0.7403 for K-S Test and t-test), indicating that, even at gridpoint scale, PGW_T_ZG_gp's precipitation and

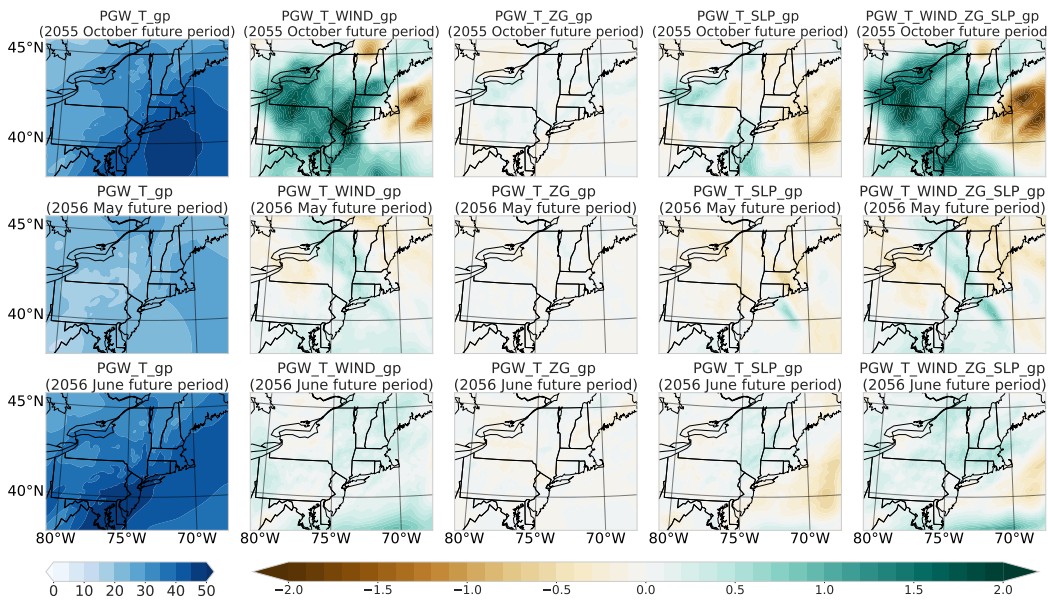

**Figure 11.** Period mean precipitable water (mm) over the inner domain from the simulation with temperature perturbation at each gridpoint, and the differences between the simulations with additional perturbations of wind, geopotential height, sea level pressure and their combination at each gridpoint and the simulation with temperature perturbation only.

2-meter temperature have the same distribution and mean values as PGW_T_gp. This lends evidence to our claim that modifying ZG (geopotential height) in PGW simulations makes little difference to the final simulation and is not necessary. Also, the statistical tests further confirm that different PGW methods will impact the simulated weather events at the gridpoint scale (the daily precipitation and 2-meter temperature at each gridpoint) but have smaller impacts on the regional mean meteorological fields (the regional mean precipitation and 2-meter temperature).

This study focuses on examining the impacts of different perturbation modifications on PGW simulations during three storm events. Note that the climate perturbations in PGW simulations will be slightly different from the original climate perturbations (from GCMs) because in the WRF runs, adjustment of atmospheric circulations and differences between WRF and GCMs (e.g., parameterizations, model resolutions) can alter the original climate perturbations to some degree. Even the modifications on the thermodynamic fields can alter the dynamic fields through their interactions (Dougherty and Rasmussen, 2021). These mechanisms also make different storms exhibit different sensitivity to the choice of the PGW method. And that's why we find that the 2005 Oct Flood, which is caused by a cold frontal system, is more sensitive to the climate perturbations applied at the boundary. In this study, we directly employ the climate perturbations from CESM1 and do not explore their underlying reasons (e.g., why the surface onshore wind is projected to increase). As one of few PGW studies focusing on the PGW methods, we do not attempt to be comprehensive and there are several shortcomings to note. For example, due to the large number of simulations we have to run and the limited computational resources, we chose a relatively coarse resolution along with a

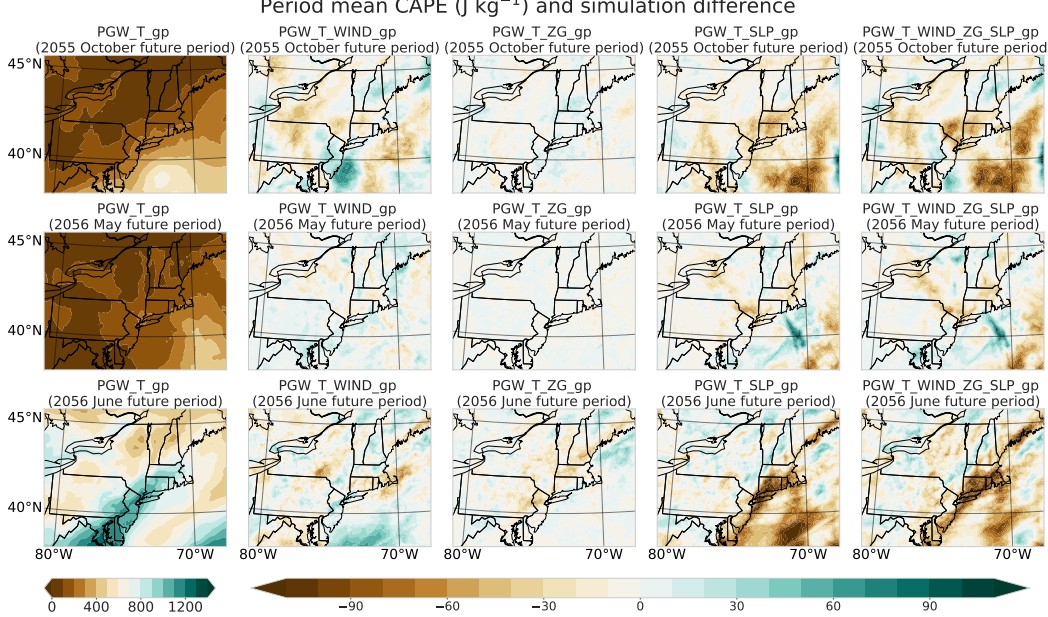

**Figure 12.** Period mean convective available potential energy (CAPE) (J kg⁻¹) over the whole inner from the simulation with temperature perturbation at each gridpoint, and the differences between the simulations with additional perturbations of wind, geopotential height, sea level pressure and their combination at each gridpoint and the simulation with temperature perturbation only.

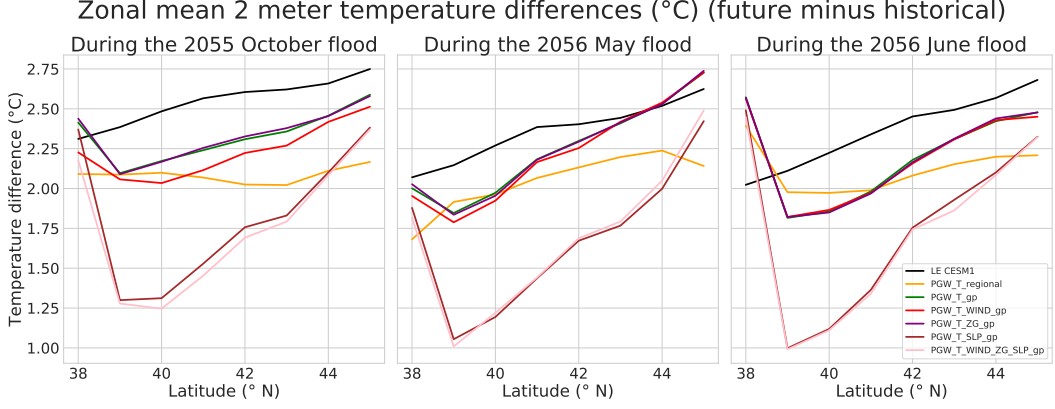

**Figure 13.** Differences of zonal mean 2-meter temperature (°C) over the inner domain between each PGW simulation of the future and the historical simulation averaged over the flood periods.

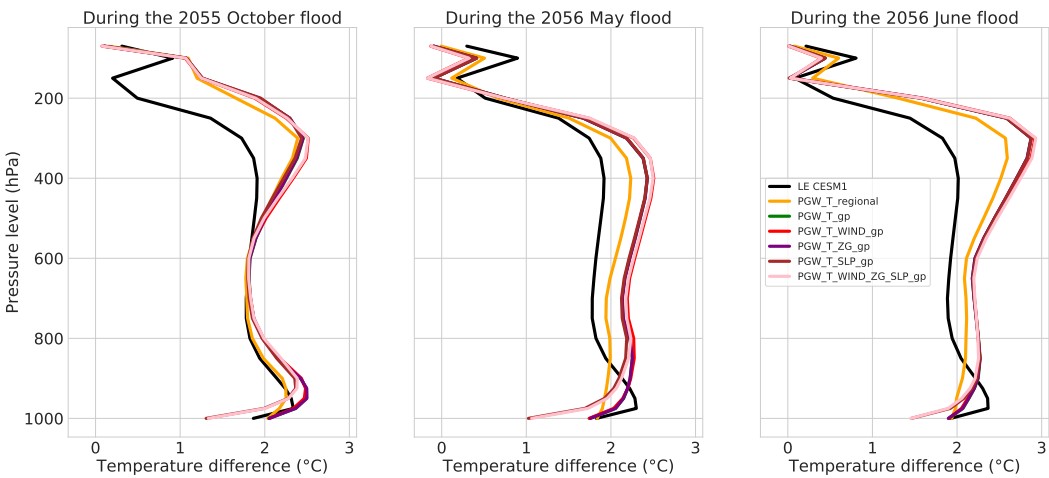

**Figure 14.** Regional mean temperature differences (°C) at each pressure level over the inner domain between each PGW simulation of the future and the historical simulation averaged over the flood period.

cumulus scheme (the analogous setting has been used in our previous studies (Ullrich et al., 2018; Xue and Ullrich, 2021b)), which can degrade the simulation performance; however, as shown in the model validation, our simulations perform well with

the help of spectral nudging. As the physics parameterizations and land model play important roles in simulating the storms and their interactions with the environment, our results on the sensitivity of the simulations to the PGW modifications may depend on the specific parameterizations and land model used. Also, the number of flood events and study regions is limited. Rather than aiming at an exhaustive investigation of the sensitivity of the PGW method to experiment design, our goal is to bring awareness to the impacts PGW choices on the conclusions drawn from PGW studies. We expect additional work is needed

to investigate the sensitivities highlighted in this study in other contexts (e.g., in inland areas or for other forms of extreme weather).

## 5 Conclusions

Although there exist discrepancies among experiments with different PGW methods, the most significant gridpoint-level differences emerge primarily from the displacement of storm events. This suggests that one should take care in projecting future

changes to precipitation amount and related fields at specific grid points within the domain. If we look at the regional scale effects, relative differences between different PGW experiments are less than 10%. This study thus supports the robustness of the PGW method for projecting regional-scale changes. With that said, some notable differences suggest greater sensitivity of particular events to the experiment design (specifically, the perturbation modification): in particular, the 2005 October storm appears more sensitive to the choice of meteorological perturbations than the other two events. This is unsurprising, as some

storm events are more sensitive to small changes in their environment than other storm events, which can result in significant

changes in their subsequent behavior. The 2005 October event was driven by a passing cold front, while the other two events were related to localized lows. Our results suggest that storms driven by frontal systems and stronger synoptic forcing may be more sensitive to the PGW experimental design because the climate perturbations may change the storm events more systematically. However, regardless of the underlying drivers of a particular event, at the regional mean scale, the differences among PGW experiments tend to be relatively small.

Nonetheless, when choosing a particular PGW experiment design, one needs to be aware of the consequences of the particular design decisions being made. In this study, we found that some common assumptions do not hold, and unrealistic results can emerge when choosing to perturb certain meteorological variables. In particular, we find that employing a regionally uniform temperature perturbation does not prevent the generation of spurious waves in the simulation; indeed, essentially all of our experiments did produce (climatologically insignificant) spurious waves during strong weather events. Additionally, we find that modifying the SLP in WRF leads to an underestimation of the warming signal and produces stronger wave-like noises. These limitations make PGW_T_regional and PGW_T_SLP_gp less favorable options for PGW experiments.

Within this study we are able to draw the following conclusions about sensitivity to PGW experimental design:

(1) The PGW experiment with regional mean temperature perturbation produced much higher precipitation totals than any experiment with gridpoint perturbations because such experiment design does not preserve the warming contrast between land and ocean, which is a robust regional feature of global warming.

(2) For experiments at the gridpoint scale, the application of geopotential height perturbations does not appreciably modify the simulation in WRF. This is because this field is already adjusted by the WRF Preprocessing System using the modified temperature fields.

(3) Gridpoint application of wind perturbations can appreciably impact the simulated precipitation because of the effect these perturbations have on vapor transport. Since the NEUS primarily receives precipitation from the southwest and on-shore southwesterly winds are enhanced under future climate change, our experiments with wind perturbations generally show enhanced total overland precipitation during flood periods.

(4) Sea level pressure perturbations directly affect the temperature field, producing cooling where SLP is enhanced and warming where SLP is reduced. Since the SLP increase in our simulations is primarily over the ocean, these perturbations in turn reduce specific humidity and subsequently reduce overland precipitation. SLP adjustments also affect CAPE and lead to a reduction in convection and convective precipitation.

Based on the results of our analysis, we recommend perturbation of both temperature and wind at the gridpoint scale for future WRF-based PGW studies. This recommendation captures the spatially-dependent impacts of climate change on both thermodynamic and dynamic fields, without modifying unnecessary fields (i.e., geopotential height). Modifying both variables should (in theory) also reduce gravity wave noise, although we did not find an appreciable change in the presence of this noise when wind perturbations were included. An alternative is to only include temperature perturbations at the gridpoint scale,

which would enable isolation of the thermodynamical effects of climate change, and avoidance of the uncertain impact of climate change on dynamic fields.

*Data availability.* The WRF model simulation data mentioned in this paper is available from ZENODO at 10.5281/zenodo.6609204.

*Author contributions.* Paul Ullrich and Zeyu Xue conceptualized this study and completed the manuscript. L. Ruby Leung completed the manuscript and revision. Zeyu conducted corresponding simulations on NERSC CORI.

*Competing interests.* The authors declare that they have no conflict of interest.

*Acknowledgements.* This research was supported by the RGMA and MSD program areas in the U.S. Department of Energy's Office of Biological and Environmental Research as part of the multi-program, collaborative Integrated Coastal Modeling (ICoM) project with Contract No. DE-AC05-76RL01830. This project is also supported by the National Institute of Food and Agriculture, U.S. Department of Agriculture, hatch project under California Agricultural Experiment Station project accession no. 1016611. This research used resources of the National Energy Research Scientific Computing Center (NERSC), a U.S. Department of Energy Office of Science User Facility located at Lawrence Berkeley National Laboratory, operated under Contract No. DE-AC02-05CH11231 using NERSC award BER-ERCAP0020801. The authors certify that they have no affiliations with or involvement in any organization or entity with any financial interest, or non-financial interest in the subject matter or materials discussed in this manuscript.

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
