# Peer review of "Sensitivity of the pseudo-global warming method under flood conditions: A case study from the Northeastern U.S."

_EGUsphere, 2022_

## Author Comment (AC3)

**Response letter to Referee Reviewer 1**

**REVIEWER COMMENTS**

**Referee Reviewer 1**

*This study evaluates the sensitivity of the PGW method to regionally uniform vs. gridpoint scale perturbations and the inclusion of different perturbed meteorological variables. The authors study the results in the context of 3 different flood events that occurred in the Northeastern U.S. using WRF-simulations forced by ERA5 and ERA5 perturbed with CESM-LENS.*

*I thought the basis of this study was interesting, given that there is little consistency in how PGW simulations are designed, but the lack of clarity in the methods, the lack of including key PGW literature, and arguments that did not make sense resulted in a paper that had too many issues to make a strong argument for what the authors were trying to show. For this reason, this paper cannot be accepted unless substantial revision is undergone. I will provide more specific comments below to explain my rationale.*

Dear reviewer, thank you very much for your questions and suggestions. We admit this paper had some issues you mentioned, but we think most issues are not fundamental and are caused by misunderstanding and lack of enough clarification. Therefore, we have responded to all issues you mentioned and hope this response addresses your concerns. Corresponding revisions have been made to the paper.

**General:**

*1. I am confused about how you evaluate the influence of different meteorological variables in your paper for each flood event. The figure captions merely state "2055 October", "2056 May" and "2056 June". However, each flood event corresponding to those months only lasts a few days of the month. So, when you present your Figures 9-13, are these fields being averaged over the timeframe of the month that contains each flood or is it averaged over the time period of the flood? This must be clearly stated because it muddles the interpretation of your figures.*

*I am concerned that if you take the mean over the month in which the flood occurs in the future simulation, the changes in fields like pressure and temperature are not just a result of the future perturbation, but are also being modified by the mesoscale processes within the storm-producing flood (and any other precipitation events that occurred during that month).*

*If these figures are just averaged over the time period of the flood (including when it is raining), again, you are introducing mesoscale modifications of the perturbations, including mesolows, outflows, and cool pools. It is impossible to evaluate these figures and what the changes in pressure, temperature, and CAPE mean if they are already being contaminated by the presence of storms.*

Sorry for the misunderstanding and confusion. On line 99-100, we state "This period is chosen as it includes the three major flood events in Table 1." And, as we defined in Table 1, the 2005 Oct Flood, 2006 May Flood, and 2005 June flood periods refer to the Northeastern U.S. flooding of October 2005, the New England Flood of May 2006, and the 2006 Mid-Atlantic United States flood, which has the duration from October 7th to 17th 2005, May 12th to May 20th, 2006 and June 23rd to July 5th, 2006. Therefore, in all analysis and corresponding figures (for example Figures 9-13), the "2055 October", "2056 May" and "2056 June" refer to the period of October 7th to 17th 2005, May 12th to May 20th, 2006 and June 23rd to July 5th, 2006, respectively. The period mean refers to the average of these three periods. We will add clarification in the manuscript.

We do average over the flood period, and it's no doubt that the climate perturbations in the WRF simulation will be different from the original climate perturbation (from GCMs) we added to the PGW runs. In the WRF simulations, there always exist atmospheric phenomena and circulations that can alter the original climate perturbations to some degree. In general, we do want to include the impacts of different storms on the climate perturbations, which makes different floods exhibit different sensitivity to the choice of PGW method. That's why we studied three different storm events and concluded that the 2005 Oct Flood, which is caused by a cold frontal system, is more sensitive to the climate perturbations applied at the boundary.

More importantly, we can confirm that the occurrence of the storms will not significantly change the impacts of sea level pressure perturbation on surface air temperature. For example, in Figure R1, we plot the simulated monthly mean 2 meter temperature difference between PGW_T_gp and PGW_T_SLP_gp during the months with lowest monthly mean precipitation (2056 January to March) in our simulation period. We can see that PGW_T_SLP_gp still has consistently lower 2 meter temperature over the sea. This illustrates that our conclusions regarding the SLP perturbation hold even in the absence of storms.

[Figure]

Monthly mean 2 meter temperature (°C) and simulation difference during the driest simulated months

*2. This paper lacks a complete understanding of the PGW literature and is missing some key papers, which is necessary if you are evaluating the utility of this method. Papers to include (not exhaustive):*

*Dougherty, E., and K. L. Rasmussen, 2020: "Changes in flash flood–producing storms in the United States. J. Hydrometeor., 22, 2221–2236, https://doi.org/10.1175/JHM-D-20-0014.1."*
*Dougherty, E., and K. L. Rasmussen, 2021: "Variations in flash flood-producing storm characteristics associated with changes in vertical velocity in a future climate in the Mississippi River Basin. J. Hydrometeor., 21, 671–687, https://doi.org/10.1175/JHM-D-20-0254.1.*
*Mahoney, K., D. Swales, M. J. Mueller, M. Alexander, M. Hughes, and K. Malloy, 2018: An examination of inland-penetrating atmospheric river flood event under potential future thermo-dynamic conditions. J. Climate, 31, 6281–6297, https://doi.org/ 10.1175/JCLI-D-18-0118.1.*
*Lackmann, G. M., 2013: The south-central U.S. flood of May 2010: Present and future. J. Climate, 26, 4688–4709, https://doi.org/10.1175/JCLI-D-12-00392.1.*

*Liu, C., and Coauthors, 2016: Continental-scale convection-permitting modeling of the current and future climate of North America. Climate Dyn., 49, 71–95, https://doi.org/10.1007/s00382-016-3327-9.*

*Prein, A. F, C. Liu, K. Ikeda, S. B. Trier, R. M. Rasmussen, G. J. Holland, and M. P. Clark, 2017: Increased rainfall volume from future convective storms in the US. Nat. Climate Change, 7, 880–884, https://doi.org/10.1038/s41558-017-0007-7.*

*Rasmussen, K. L., A. F. Prein, R. M. Rasmussen, K. Ikeda, and C. Liu, 2017: Changes in the convective population and thermodynamic environments inconvection-permitting regional climate simulations over the United States.*

Given constraints on space we limited our review to about 20 PGW-related papers, although we do admit that citing more papers can make this paper more comprehensive and we will cite the papers you mentioned. But the absence of these citations doesn't seem to be a fundamental flaw in the work, and does not alter our conclusions.

*3. I think the gravity wave noise section needs to be reevaluated. Previous PGW studies including Lackmann (2013) and Mahoney et al. (2018) have found that gravity wave adjustments in their studies are relatively small and only apparent during the early spinup periods. This makes me skeptical of what you are arguing, given that I do not see any evidence of gravity waves in your figures and the presence of wave-like noise in the presence of storms implies that you are seeing storm-generated gravity waves, which are physical and not just a model artifact.*

Two of the papers mentioned above (namely, Lackmann (2013) and Mahoney et al. (2018)) do mention that the gravity wave adjustments are only strong during the early stages of the simulation. But both of them only mention it in one sentence and do not provide any evidence supporting these results (such as figures or wave analysis). For example, Lackmann (2013) only state: "Any imbalance between the wind and mass field is sufficiently small to preclude strong gravity wave adjustment early in the future simulation." Mahoney et al. (2018) state: "Gravity wave adjustment between the wind and mass fields early in the simulations is accordingly short-lived." Because both of them do not provide any further explanation and figures, we cannot comment on their conclusions. We guess that they examined the mean field (like the daily mean), where the gravity wave noise is invisible, instead of examining the simulation at high temporal resolution.

In our simulations, we also find that the gravity waves are not obvious if we look at the period mean (for example as shown in Figures S6 and S7); however, if we look at the sea level pressure at high temporal resolution, as it evolves (the animation in the citation in Ln 203 and Figure R2 here, available at https://zenodo.org/record/6544880#.Y0YyQuzMJ9s ), wave-like noise during the flood period and in the PGW simulations is much stronger when compared with the historical run. Also, the wave-like noise is more significant during storm events and that's why we say "the observed noise is enhanced by strong advection during these extreme weather events, as it is much more obvious during these storm events." More importantly, as shown in Figure R3 as attached, we can see the wave-like noise in the magnitude spectrum after Fourier Transform in the PGW runs (the high-frequency signals are shown as light circles in the magnitude spectrum)

but do not observe such a signal in the historical run. This further confirms that the wave-like noise does exist in our simulation during the flood periods and that it is not obvious (or present to the same degree) in the control simulation.

Please refer to https://zenodo.org/record/6544880#.Y0YyQuzMJ9s
Figure. R2 The animation of sea level pressure during the 2005 October Flood and returned 2055 October Flood

The Sea Level Pressure Field and Its Magnitude Spectrum after Fourier Transform

[Figure]

Sea Level Pressure in Historical Run on 2005 Oct 12th 00:00

[Figure]

Magnitude Spectrum in historical Run on 2005 Oct 12th 00:00

[Figure]

Sea Level Pressure in PGW_T_gp Run on 2005 Oct 12th 00:00

[Figure]

Magnitude Spectrum in PGW_T_gp Run on 2055 Oct 12th 00:00

[Figure]

Sea Level Pressure in PGW_T_WIND_gp Run on 2005 Oct 12th 00:00

[Figure]

Magnitude Spectrum in PGW_T_WIND_gp Run on 2055 Oct 12th 00:00

Figure. R3 The sea level pressure field and its magnitude spectrum after Fourier transform

**Specific comments**

*Lines 9–11: I would be careful about drawing conclusions from the experimental design alone, as even running simulations in different versions of WRF can cause storm displacements.*

Thanks for the suggestion. We replaced "experimental design" with "perturbation modification" here to make it more accurate.

*Lines 20–23: I would recommend looking at the IPCC report and CORDEX studies to cite how confident future projections are of extreme precipitation in the Northeast, in addition to Melillo et al. 2014.*

This citation has been added as you suggested.

*Line 55: Please add Liu et al. 2016 to this list of PGW studies.*

This citation has been added.

*Lines 85–86: Please add Dougherty and Rasmussen (2020,2021), Prein et al. (2017), and Rasmussen et al. (2020) to this list.*

Corresponding citations have been added.

*Lines 89–91: I am curious why you decided not to perturb moisture in this study–can you please explain this in your methods?*

We perturb the moisture condition by fixing the relative humidity and applying the warming perturbations instead of directly changing the specific humidity. This is a common approach under the PGW method (e.g., the original PGW papers of Schär et al., 1996 and Frei et al., 1998), since an examination of future and historical relative humidity over this region suggests that changes to relative humidity are not statistically significant. If specific humidity were perturbed then a grid cell where relative humidity was originally 100% could drop below 100% leading to suppression of precipitation. Perturbing specific humidity could also lead to relative humidity being increased above 100%, which is unphysical. For these reasons specific humidity is not perturbed directly.

*Line 103: Please add Liu et al. (2016) and Beck et al. (2019) to this citation: Beck, H. E., and Coauthors, 2019: Daily evaluation of 26 precipitation datasets using Stage-IV gauge radar data for the CONUS. Hydrol. Earth Syst. Sci., 23, 207–224, https://doi.org/ 10.5194/hess-23-207-2019.*

Citations have been added.

Line 103–105: Please briefly list these parameterizations in the text.

The physical parameterizations have been added to Table S1.

*Line 105: Why is CLM rather than Noah-MP used for a land model?*
*Lines 105–107: I think the choice of the land model is actually quite important. I understand it is not central to your study, but Barlage at el. (2021) showed the warm, dry bias in the Central U.S. from PGW simulations could be reduced by adding groundwater to Noah-MP in CONUS-wide PGW simulations.*

There are a number of reasons why CLM was selected for this study: As described in the paper, both the physical parameterizations and the land model (CLM) we used are from CESM1, which is widely regarded as a high-performance climate model (Kay et al., 2015; Sillmann et al., 2013; Karmalkar et al., 2019). So, CLM is more consistent with the physical parameterizations we used. Also, the CLM model is also the most complicated and expensive of the available options in WRF, and one that shows reasonable performance across a variety of geographies (Jin et al., 2010; Case et al., 2008; Ullrich et al., 2018). Indeed, although Barlage at el. (2021) showed the warm, dry bias in the Central U.S. from PGW simulations could be reduced by adding groundwater to Noah-MP in CONUS-wide PGW simulations, it did not confirm that Noah-MP can outperform the CLM land model. More importantly, this study focuses on the dry periods over the Central U.S. but our study focuses on flood periods over NEUS.  We know that model performance varies from region and phenomenon.

*Lines 110–111: Can you explain why you chose to use 10 km for your inner domain? Many of the PGW studies use this method in order to simulate storms in a future climate at convection-permitting resolutions. At 10-km, you are not quite at convection-permitting scales so it would be helpful to understand the rational why this grid-spacing is beneficial for your study. A caveat that the cumulus parameterization is turned on would also be helpful, since this will likely greatly affect your results.*

We also expect that conducting the simulation at a finer resolution, particularly a convection-permitting resolution, will improve the simulation accuracy; however, due to the large number of simulations we have to run, we choose a relatively coarse resolution along with a cumulus scheme. While we agree this is a shortcoming of the study that is largely necessitated by our limited computational resources, we do note that it is not uncommon in the literature to perform PGW simulations at analogous resolutions (a similar setup has been employed successfully in our previous studies: Ullrich et al., 2018 and Xue and Ullrich 2021b). As the aim of this study is to examine the response of the PGW simulations to different perturbation modification methods, the simulation accuracy itself is not the primary concern in this study. This shortcoming is now discussed in the paper as a limitation in the newly added discussion section; we also hope to explore these conclusions at higher resolution in future studies.

*Lines 112–114: Can you specify the spatial scales here? Did you employ nudging through the entire vertical domain or just above the boundary layer?*

The spectral nudging uses guv, gt, gq and gph equaling to 0.0003. The xwavenum and ywavenum are equal to 3. The nudging is only applied above the boundary layer.

*Table 1: How were the start and end dates of the flood determined?*

The start and end dates of flood periods are informed by the reference papers in the Meteorological Cause column in Table 1. To get the exact dates, we define the start of the flood as the day when regional daily mean precipitation over the inner domain is larger than 1.8 mm/day, and the end of the flood as the day when two consecutive days after that day is less than 1.8 mm/day.

*Lines 138–140: Do you also vary the 2D 10-m winds, 2-m temperature in ERA5?*

*We do not modify the 2D 10-m winds but modify the 2-m temperature as LE CESM1 only provides the 10-meter wind speed project instead of the projections of u10 and v10. Modifying 2D 10-m winds (u10 and v10) is unnecessary because they are diagnostic variables and are only required in the initialization.*
*https://forum.mmm.ucar.edu/threads/forcing-surface-winds.8666/*
*https://forum.mmm.ucar.edu/threads/running-wrf-with-cesm2-data.9965/#p20292*

*Lines 150–153: Why did you decide to use the CPC precipitation? It has nice global coverage but is much coarser resolution than your 10-km model data. I would suggest comparison to something higher resolution like Stage-IV or PRISM precipitation.*

The reason to use CPC precipitation is that it's one of the most widely used and accepted observational precipitation datasets. We admit its resolution is coarser compared with our simulations and that's why we also use high-resolution and high-quality precipitation datasets – ERA5 and IMERG_V6.

*Figure 3: Did you regrid these data to all be at the same grid-spacing?*

We did not regrid the data. However, motivated by this comment, we have replotted Figure 3 with all observed/reanalysis data interpolated into the same grid as our simulations.

*Lines 165–167: This is for enhanced warming over land for gridpoint perturbations if I am interpreting Fig. 4 left column correctly, right? Please indicate that you are referring to Fig. 4 in this statement for clarity.*

Thank you for pointing it out. We have mentioned it in the paper.

*Lines 174–177: I don't see much difference in Fig. S9 or Fig. S10- can you show a difference plot to highlight this more clearly?*

Sorry for the misunderstanding. When we refer to Figures S9 and S10, we mean to say that Figures S9 and S10 show that the 2005 October Flood (2055 October Flood) has a much stronger uplift (as shown by the green areas in the first row in Fig. S10) near the coastal region and much stronger onshore wind (as shown by the red areas in the first row in Fig. S9) compared with the 2006 May Flood (2056 May Flood) and the 2006 June Flood (2056 June Flood). In Figures S9 and S10, we can easily observe it by comparing the first row with the second and third rows.

*Lines 181–182: Again, I am failing to see where precipitable water is much higher in the regional mean simulation than the gridpoint. Can you provide a domain mean in Figure S2 and put those numbers in each panel?*

To better illustrate the precipitable water difference between PGW_T_regional and PGW_T_gp, we have plotted the figure below and added it to the supplement.

[Figure]

Period precipitable water (mm d$^{-1}$) and simulation difference

Figure. R4 (Left) Period mean precipitable water (mm) over the whole domain from the simulation with temperature perturbation at regional mean scale. (Right) Difference between the period mean precipitable water (mm) from the gridpoint perturbation simulation and regional mean simulation from the left column.

*Line 203–205: When you are saying that dynamical fields are assumed to be unchanged, I think you are speaking very specifically about uniform temperature perturbations in the regional sense. However, I do want to clarify that you have to be careful about making these statements since this is not the case in all PGW simulations. I think it is more accurate to say that the PGW method assumes that the dynamical changes are much smaller order magnitude than thermodynamic changes (Liu et al. 2016; O'Gorman 2015). Furthermore, it is impossible to assume the dynamics don't change due to change in thermodynamics (i.e., temperature), because these interact with each other. For example, Dougherty and Rasmussen (2021) show that storms with stronger updrafts show greater increases in rainfall in 4-km PGW simulations, likely due to the latent heat feedback suggested by Trenberth (1999).*

Sorry for the misunderstanding. By saying that dynamical fields are assumed to be unchanged, we mean that the dynamical field in the WRF input is not changed. Certainly, the modified thermodynamic fields will change the original dynamical fields as a result of mechanisms like the geostrophic adjustment. We will clarify this problem in the newly added discussion section and cite the reference you mentioned.

*Lines 209–210: How do you know PGW_T_regional is overestimating precipitation when there is no future "truth" or observations to compare it to? What is your baseline for this statement?*

While we cannot know for sure if PGW_T_regional overestimates the precipitation, we mean to say that PGW_T_regional overestimates precipitation compared with the gridpoint perturbation methods. However, there are good reasons to believe that PGW_T_regional is behaving unphysically. Because the PGW_T_gp employs the time-varying temperature perturbations at each gridpoint and pressure level (3D space + 1D time), it can capture the horizontal spatial variance of the temperature perturbation. But PGW_T_regional only uses the time-varying temperature perturbations at the regional mean scale at each pressure level (2D space + 1D time), and so does not include any information about the horizontal spatial variance of the temperature perturbation. There is ample evidence to indicate that the temperature perturbation over the land will be larger than the temperature perturbation over the sea, and so it is likely that the regional mean modification leads to future temperatures over the sea that are higher than climate model projections indicate. The overestimated temperature perturbation in PGW_T_regional can be seen in Figure 4 (in the main paper). Since the relative humidity is fixed in our PGW runs, the change of specific humidity is determined by the temperature perturbation, and is subsequently larger in the regional mean simulation. Since precipitation in this region is largely driven by on-shore flow from the Atlantic, the enhanced specific humidity leads to enhanced vapor transport and increased moisture convergence (and subsequently higher precipitation in Figure 5 and 6 of

the paper). Therefore, we claim for this choice of domain, the simulated precipitation in PGW_T_gp is more reliable compared with PGW_T_regional.

*Line 226: Are you referring to the purple cross in this figure? If so, please state it as such.*

Yes, the enhanced precipitation refers to the purple cross in Figure 6 and we have stated it.

*Line 229: Is the reduction over the sea compared to the historical simulation or T-only experiment?*

It's compared with PGW_T_gp and we have clarified it in the paper.

*Lines 232–233: I think it's important to think about the larger view of why the wind magnitudes are stronger. The other two floods are warm-season events in May and June that have weaker synoptic forcing, whereas in October, synoptic dynamics and the forcing from it are quite strong. This would be my guess as to why the wind perturbation is stronger in the October flood than the others, and I think you should note that somewhere in your discussion*

We agree that understanding why the onshore wind magnitude increase is stronger in October is important; however, in this case it's mainly determined by the wind projection and corresponding perturbation from LE CESM1. Because this study focuses on how will the PGW simulations respond to different perturbation modification methods, explaining what causes the wind projections in GCMs are out of the scope of this study. Other papers (for example, Kulkarni et al., 2014) also show that the surface wind speed is projected to increase more in the winter over NEUS in GCMs' projections. We will briefly talk about it in the discussion.

*Line 235: Again, please be clear which marker you are referring to in Figure 6 when discussing these results.*

"Referring to the difference between the purple cross and green triangle in Fig. 6" has been added.

*Lines 262–263: That's not true- hypsometric equation states that a thicker layer = a warmer layer. However, by ideal gas law, $p = \rho RT$, pressure and temperature are proportional such that higher pressure = warmer temperature.*

As we stated, WRF uses a hybrid vertical coordinate which is of roughly constant layer thickness at ground level, and so near-surface changes to SLP are incorporated at constant volume. In hypsometric equation ($Z_2 - Z_1 = \frac{R \times T_v}{g} \times \ln(\frac{P1}{P2})$), when $Z_2 - Z_1$ is fixed, a increasing $P_1$ indicates a decreasing $T_v$ (mean virtual temperature). The ideal gas law describes the state of a hypothetical ideal gas (air parcel), and it's not applicable here because, although density varies inversely with layer thickness, air density is not immediately known.

*Line 266–267: The colder temperatures and reduced precipitable water appears to be*

*true only over the sea surface.*

Note that the precipitable water is defined as the depth of water in a column of the atmosphere instead of precipitable water in the surface atmosphere, and as we show in Figure 12 that there is a generally reduced precipitable water over the sea region instead of the surface of sea. In fact, most of the atmospheric moisture and precipitable water is present at the near surface. Therefore, a decrease in surface temperatures will cause the decrease in total precipitable water.

*Line 27–274: Were the wave-like noises found only in the paper you cite? Because I do not see any in Figure S6 or S7.*

Please refer to our response to the general issue 3. Figures S6 and S7 are the period mean plots and so do not exhibit the wave-like noise as this noise is not stationary; however, if we look at the animation of sea level pressure during flood periods (https://zenodo.org/record/6544880#.Y0YyQuzMJ9s), we can clearly observe the wave-like noise.

*Figure 14: Is this just the 2056 May mean temperature from LE CESM1 without running any PGW simulations?*

Yes, the black line illustrates the period mean temperature difference between historical and future periods projected by the multi-ensemble mean of LE CESM.

*Line 290–292: I would add "stronger synoptically-forced systems" after frontal systems.*

This has been added.

**Technical comments:**

*Table 1: Please replace "huge" with "large", since the former is too colloquial.*

Corrections have been made.

*Figure 2:*
*The top left panel appears to have a different color scheme than the rest of the figure. Please fix this.*
*Caption: Is the average temperature the average monthly temperature? If so, please include that in the figure caption.*

Thank you greatly for pointing it out. As suggested by the reviewer 2, we plotted the temperature difference in the second and third columns in Figure 2. The average temperature is the period mean temperature during three flood periods (the periods we defined the Table 1). We have replaced the average temperature with period mean temperature in the caption.

*Lines 170–172: This is confusing as written. I would suggest rewriting to say "increase*

*from 12.69 mm/d to 13.86 mm/d in Oct 2005, 7.19 to 7.83 mm/d in May 2006, and 7.43 to 7.88 mm/d in June 2006)".*

We have modified the sentence as you suggested.

*Figure 4:*
*I would change the label of the left-column figures to PGW_T_gp - PGW_T_regional since you are showing a difference.*
*Please flip the colorbar so it matches the intuition that red is warmer and blue is cooler.*

In all period mean and simulation difference figures (like Figures 2, 3, 5, 8 and 9), the left column shows the baseline simulation, while the rest of the columns indicate the differences compared with the baseline (as stated in the captions). Otherwise, the subtitles might be too long. For example, for the difference between *PGW_T_WIND_SLP_gp and PGW_T_gp, the subtitle will be PGW_T_WIND_SLP_gp - PGW_T_gp which is lengthy and redundant.*

Thanks for your suggestion on color bar. We have reversed it.

*Figure 7 should be the last figure given that the first time it is mentioned is on Line 278.*

We have relocated it to the correct position.

*Figure 10: I think you mean hPa instead of "(mm/d)" in the figure caption, right?*

We have corrected the caption.

**Reference**

Schär, C., Frei, C., Lüthi, D., and Davies, H. C.: Surrogate climate-change scenarios for regional climate models, Geophysical Research Letters, 23, 669–672, 1996.

Frei, C., Schär, C., Lüthi, D., and Davies, H. C.: Heavy precipitation processes in a warmer climate, Geophysical Research Letters, 25, 1431–1434, 1998.

Karmalkar, A. V., Thibeault, J. M., Bryan, A. M., and Seth, A.: Identifying credible and diverse GCMs for regional climate change studies-case study: Northeastern United States, Climatic Change, 154, 367–386, 2019.

Sillmann, J., Kharin, V., Zhang, X., Zwiers, F., and Bronaugh, D.: Climate extremes indices in the CMIP5 multimodel ensemble: Part 1. Model evaluation in the present climate, Journal of Geophysical Research: Atmospheres, 118, 1716–1733, 2013.

Kay, J. E., Deser, C., Phillips, A., Mai, A., Hannay, C., Strand, G., Arblaster, J. M., Bates, S., Danabasoglu, G., Edwards, J., et al.: The Community Earth System Model (CESM) large ensemble project: A community resource for studying climate change in the presence of

internal climate variability, Bulletin of the American Meteorological Society, 96, 1333–1349, 2015.

Ullrich, P., Xu, Z., Rhoades, A., Dettinger, M., Mount, J., Jones, A., & Vahmani, P. (2018). California's drought of the future: A midcentury recreation of the exceptional conditions of 2012–2017. Earth's Future, 6 (11), 1568–1587.

Jin, J., Miller, N. L., & Schlegel, N. (2010). Sensitivity study of four land surface schemes in the WRF model. Advances in Meteorology, 2010.

Case, J. L., Crosson, W. L., Kumar, S. V., Lapenta, W. M., & Peters-Lidard, C. D. (2008). Impacts of high-resolution land surface initialization on regional sensible weather forecasts from the WRF model. Journal of Hydrometeorology, 9 (6), 1249–1266.

Sujay Kulkarni, Huei-Ping Huang, "Changes in Surface Wind Speed over North America from CMIP5 Model Projections and Implications for Wind Energy", Advances in Meteorology, vol. 2014, Article ID 292768, 10 pages, 2014. https://doi.org/10.1155/2014/292768

---

## Author Response (AR1)

**Response letter to Referee Reviewers**

**Table of Contents**

**REVIEWER COMMENTS - Referee Reviewer 1**

*This study evaluates the sensitivity of the PGW method to regionally uniform vs. gridpoint scale perturbations and the inclusion of different perturbed meteorological variables. The authors study the results in the context of 3 different flood events that occurred in the Northeastern U.S. using WRF-simulations forced by ERA5 and ERA5 perturbed with CESM-LENS.*

*I thought the basis of this study was interesting, given that there is little consistency in how PGW simulations are designed, but the lack of clarity in the methods, the lack of including key PGW literature, and arguments that did not make sense resulted in a paper that had too many issues to make a strong argument for what the authors were trying to show. For this reason, this paper cannot be accepted unless substantial revision is undergone. I will provide more specific comments below to explain my rationale.*

We thank the reviewer for the questions and suggestions that help improve the manuscript. We have responded to all the issues raised and hope this response addresses all the concerns. Corresponding revisions have been made to the paper.

**General:**

*1. I am confused about how you evaluate the influence of different meteorological variables in your paper for each flood event. The figure captions merely state "2055 October", "2056 May" and "2056 June". However, each flood event corresponding to those months only lasts a few days of the month. So, when you present your Figures 9-13, are these fields being averaged over the timeframe of the month that contains each flood or is it averaged over the time period of the flood? This must be clearly stated because it muddles the interpretation of your figures.*

*I am concerned that if you take the mean over the month in which the flood occurs in the future simulation, the changes in fields like pressure and temperature are not just a result of the future perturbation, but are also being modified by the mesoscale processes within the storm-producing flood (and any other precipitation events that occurred during that month).*

*If these figures are just averaged over the time period of the flood (including when it is raining), again, you are introducing mesoscale modifications of the perturbations, including mesolows, outflows, and cool pools. It is impossible to evaluate these figures and what the changes in pressure, temperature, and CAPE mean if they are already being contaminated by the presence of storms.*

Sorry for the confusion with the figure captions. On line 99-100, we state "This period is chosen as it includes the three major flood events in Table 1." And, as we defined in Table 1, the 2005 Oct Flood, 2006 May Flood, and 2005 June flood periods refer to the Northeastern U.S. flooding of October 2005, the New England Flood of May 2006, and the 2006 Mid-Atlantic United States flood, which has the duration from October 7th to 17th 2005, May 12th to May 20th, 2006 and June 23rd to July 5th, 2006. Therefore, in all analysis and corresponding figures (for example Figures 9-13), "2055 October", "2056 May" and "2056 June" refer to the period of October 7th

to 17th 2005, May 12th to May 20th, 2006 and June 23rd to July 5th, 2006, respectively, in the PGW experiments for the perturbed climate of 2055/2056. The period mean refers to the average of the three flood periods within each month. We have added clarification in the manuscript.

We do show the simulations averaged over the flood period since our focus for the PGW experiments is to project how extreme events may change in the future. Our goal is to compare the results from different PGW experiments that include perturbations of different variables (T, U/V, geopotential height, sea level pressure) to determine the sensitivity of the PGW simulations to the variables perturbed. Surely the differences between the PGW experiments reflect not only the differences in the perturbations added to the initial and boundary conditions because the storms in the simulations are affected by the perturbations, which would in turn modify the atmospheric states including T, U/V, geopotential height, and sea level pressure. Hence the differences between the PGW simulations will be different from the original climate perturbations (from GCMs) added to initial and boundary conditions. Furthermore, the differences between the PGW experiments would depend on the storms being simulated because as noted above, the storms can provide feedback on the atmospheric states. This is why we studied three different storm events and concluded that the 2005 Oct Flood, which is caused by a cold frontal system, is more sensitive to the climate perturbations applied at the boundary.

More importantly, we confirm that the occurrence of the storms will not significantly change the impacts of sea level pressure perturbation on surface air temperature. For example, in Figure R1, we plot the simulated monthly mean 2-meter temperature difference between PGW_T_gp and PGW_T_SLP_gp during the months with lowest monthly mean precipitation (2056 January to March) in our simulation period. We can see that PGW_T_SLP_gp still has consistently lower 2-meter temperature over the sea. This illustrates that our conclusions regarding the SLP perturbation hold even in the absence of storms.

[Figure]

**Figure. R1 Monthly mean 2 meter temperature (°C) and simulation difference during the driest simulated months**

*2. This paper lacks a complete understanding of the PGW literature and is missing some key papers, which is necessary if you are evaluating the utility of this method. Papers to include (not exhaustive):*

*Dougherty, E., and K. L. Rasmussen, 2020: "Changes in flash flood–producing storms in the United States. J. Hydrometeor., 22, 2221–2236, https://doi.org/10.1175/JHM-D-20-0014.1."*
*Dougherty, E., and K. L. Rasmussen, 2021: "Variations in flash flood-producing storm characteristics associated with changes in vertical velocity in a future climate in the Mississippi River Basin. J. Hydrometeor., 21, 671–687, https://doi.org/10.1175/JHM-D-20-0254.1.*
*Mahoney, K., D. Swales, M. J. Mueller, M. Alexander, M. Hughes, and K. Malloy, 2018: An examination of inland-penetrating atmospheric river flood event under potential future thermo-dynamic conditions. J. Climate, 31, 6281–6297, https://doi.org/ 10.1175/JCLI-D-18-0118.1.*
*Lackmann, G. M., 2013: The south-central U.S. flood of May 2010: Present and future. J. Climate, 26, 4688–4709, https://doi.org/10.1175/JCLI-D-12-00392.1.*

*Liu, C., and Coauthors, 2016: Continental-scale convection-permitting modeling of the current and future climate of North America. Climate Dyn., 49, 71–95, https://doi.org/10.1007/s00382-016-3327-9.*
*Prein, A. F, C. Liu, K. Ikeda, S. B. Trier, R. M. Rasmussen, G. J. Holland, and M. P. Clark, 2017: Increased rainfall volume from future convective storms in the US. Nat. Climate Change, 7, 880–884, https://doi.org/10.1038/s41558-017-0007-7.*
*Rasmussen, K. L., A. F. Prein, R. M. Rasmussen, K. Ikeda, and C. Liu, 2017: Changes in the convective population and thermodynamic environments inconvection-permitting regional climate simulations over the United States.*

We thank the reviewer for providing a list of relevant papers. These papers provide important context for our study and support the gaps we identified and intended to address. We have cited these papers, along with ~20 other PGW-related papers already cited, in the revised manuscript.

*3. I think the gravity wave noise section needs to be reevaluated. Previous PGW studies including Lackmann (2013) and Mahoney et al. (2018) have found that gravity wave adjustments in their studies are relatively small and only apparent during the early spinup periods. This makes me skeptical of what you are arguing, given that I do not see any evidence of gravity waves in your figures and the presence of wave-like noise in the presence of storms implies that you are seeing storm-generated gravity waves, which are physical and not just a model artifact.*

Two of the papers mentioned above (namely, Lackmann (2013) and Mahoney et al. (2018)) do mention that the gravity wave adjustments are only strong during the early stages of the simulation. For example, Lackmann (2013) stated: "Any imbalance between the wind and mass field is sufficiently small to preclude strong gravity wave adjustment early in the future simulation." Mahoney et al. (2018) stated: "Gravity wave adjustment between the wind and mass fields early in the simulations is accordingly short-lived." Because these papers do not provide further explanations and figures, we cannot comment on their conclusions. We want to note that gravity wave noise is invisible in the mean field like daily mean, and it's not clear if the authors examined their simulations at high temporal resolution.

In our simulations, we find that the gravity waves are not obvious in the period mean (for example as shown in Figures S6 and S7). However, if we look at the sea level pressure at high temporal resolution (e.g., the animation cited in Ln 203 and Figure R3 below, available at https://zenodo.org/record/6544880#.Y0YyQuzMJ9s ), wave-like noise during the flood period in the PGW simulations is much stronger compared to the historical run. Also, the wave-like noise is more significant during storm events so we noted that "the observed noise is enhanced by strong advection during these extreme weather events, as it is much more obvious during these storm events." More importantly, as shown in Figure R3, we can see the wave-like noise in the magnitude spectrum after applying Fourier Transform to the PGW runs (high-frequency signals are shown as the light circles in the magnitude spectrum) but we do not observe such a signal in the historical run. This further confirms that the wave-like noise does exist in our simulation during the flood periods and that it is not obvious (or present to the same degree) in the historical simulation.

**Please refer to https://zenodo.org/record/6544880#.Y0YyQuzMJ9s**

**Figure. R2 The animation of sea level pressure during the 2005 October Flood and returned 2055 October Flood**

The Sea Level Pressure Field and Its Magnitude Spectrum after Fourier Transform

Sea Level Pressure in Historical Run on 2005 Oct 12th 00:00

[Figure]

Magnitude Spectrum in historical Run on 2005 Oct 12th 00:00

[Figure]

Sea Level Pressure in PGW_T_gp Run on 2005 Oct 12th 00:00

[Figure]

Magnitude Spectrum in PGW_T_gp Run on 2055 Oct 12th 00:00

[Figure]

Sea Level Pressure in PGW_T_WIND_gp Run on 2005 Oct 12th 00:00

[Figure]

Magnitude Spectrum in PGW_T_WIND_gp Run on 2055 Oct 12th 00:00

[Figure]

**Figure. R3 The sea level pressure field and its magnitude spectrum after Fourier transform**

**Specific comments**

*Lines 9–11: I would be careful about drawing conclusions from the experimental design alone, as even running simulations in different versions of WRF can cause storm displacements.*

Thanks for the suggestion. We replaced "experimental design" with "perturbation modification" here to make it more accurate.

*Lines 20–23: I would recommend looking at the IPCC report and CORDEX studies to cite how confident future projections are of extreme precipitation in the Northeast, in addition to Melillo et al. 2014.*

This citation has been added as you suggested.

*Line 55: Please add Liu et al. 2016 to this list of PGW studies.*

This citation has been added.

*Lines 85–86: Please add Dougherty and Rasmussen (2020,2021), Prein et al. (2017), and Rasmussen et al. (2020) to this list.*

Corresponding citations have been added.

*Lines 89–91: I am curious why you decided not to perturb moisture in this study–can you please explain this in your methods?*

We perturb the moisture condition by fixing the relative humidity and applying the warming perturbations to calculate the specific humidity instead of directly perturbing the specific humidity based on the GCM simulations. This is a common approach under the PGW method (e.g., the original PGW papers of Schär et al., 1996 and Frei et al., 1998), since an examination of future and historical relative humidity over the study region suggests that the relative humidity changes are not statistically significant. Perturbing specific humidity could also lead to relative humidity > 100%, which is unphysical. For these reasons specific humidity is not perturbed directly.

*Line 103: Please add Liu et al. (2016) and Beck et al. (2019) to this citation: Beck, H. E., and Coauthors, 2019: Daily evaluation of 26 precipitation datasets using Stage-IV gauge radar data for the CONUS. Hydrol. Earth Syst. Sci., 23, 207–224, https://doi.org/ 10.5194/hess-23-207-2019.*

Citations have been added.

Line 103–105: Please briefly list these parameterizations in the text.

The physics parameterizations have been added to Table S1.

*Line 105: Why is CLM rather than Noah-MP used for a land model?*
*Lines 105–107: I think the choice of the land model is actually quite important. I understand it is not central to your study, but Barlage at el. (2021) showed the warm, dry bias in the Central U.S. from PGW simulations could be reduced by adding groundwater to Noah-MP in CONUS-wide PGW simulations.*

There are a number of reasons why CLM was selected for this study: As described in the paper, both the physics parameterizations and the land model (CLM) we used in WRF are from CESM1, which is widely regarded as a high-performant climate model (Kay et al., 2015; Sillmann et al., 2013; Karmalkar et al., 2019). Additionally, using similar physics parameterizations and land model in the regional and global models help improve model consistency. We agree that the choice of the land model is quite important, but since the focus of this study is to evaluate the sensitivity of the PGW method to specific aspects of how PGW is implemented, we consider the choice of the physics parameterizations and land model as secondary to the specifics of the PGW approach. We have added a discussion section to mention some limitations of this study, including possible dependence of the sensitivity results to the specific physics parameterizations and land model used.

*Lines 110–111: Can you explain why you chose to use 10 km for your inner domain? Many of the PGW studies use this method in order to simulate storms in a future climate at convection-permitting resolutions. At 10-km, you are not quite at convection-permitting scales so it would be helpful to understand the rational why this grid-spacing is beneficial for your study. A caveat that the cumulus parameterization is turned on would also be helpful, since this will likely greatly affect your results.*

We also expect that conducting the simulation at a finer resolution, particularly at convection-permitting resolution, will improve the simulation accuracy; however, due to the large number of simulations we have to run, we choose a relatively coarse resolution along with a cumulus scheme. While we agree this is a shortcoming of the study that is largely necessitated by our limited computational resources, we do note that it is not uncommon in the literature to perform PGW simulations at analogous resolutions (a similar setup has been employed successfully in our previous studies: Ullrich et al., 2018 and Xue and Ullrich 2021b). Also, because we use the spectral nudging and suitable parameterizations, our simulations perform well as shown in Figure 2 and 3. As the aim of this study is to examine the response of the PGW simulations to different perturbation modification methods, the simulation accuracy itself is not the primary concern in this study. This shortcoming is now discussed in the paper in the newly added discussion section; we also hope to explore these conclusions at higher resolution in future studies.

*Lines 112–114: Can you specify the spatial scales here? Did you employ nudging through the entire vertical domain or just above the boundary layer?*

The spectral nudging uses guv, gt, gq and gph equal to 0.0003. The xwavenum and ywavenum are equal to 3. The nudging is only applied above the boundary layer.

*Table 1: How were the start and end dates of the flood determined?*

The start and end dates of flood periods are informed by the reference papers in the Meteorological Cause column in Table 1. To get the exact dates, we define the start of the flood as the day when regional daily mean precipitation over the inner domain is larger than 1.8 mm/day, and the end of the flood as the day when two consecutive days after that day is less than 1.8 mm/day.

*Lines 138–140: Do you also vary the 2D 10-m winds, 2-m temperature in ERA5?*

We do not modify the 2D 10-m winds but modify the 2-m temperature as CESM1 LE only provides the 10-meter wind speed project instead of the projections of u10 and v10 for both historical and future simulations. Modifying 2D 10-m winds (u10 and v10) is unnecessary because they are diagnostic variables and are only required in the initialization. Considering the length of our simulation and spin-up period, the impacts of 2D 10-m winds are neglectable. https://forum.mmm.ucar.edu/threads/forcing-surface-winds.8666/ https://forum.mmm.ucar.edu/threads/running-wrf-with-cesm2-data.9965/#p20292

*Lines 150–153: Why did you decide to use the CPC precipitation? It has nice global coverage but is much coarser resolution than your 10-km model data. I would suggest comparison to something higher resolution like Stage-IV or PRISM precipitation.*

We use CPC precipitation because it's one of the most widely used and accepted observational precipitation datasets. We admit its resolution is coarser compared with our simulations and that's why we also use high-resolution and high-quality precipitation datasets – ERA5 and IMERG_V6.

*Figure 3: Did you regrid these data to all be at the same grid-spacing?*

We did not regrid the data.  However, motivated by this comment, we have replotted Figure 3 with all observed/reanalysis data interpolated to the same grid as our simulations.

*Lines 165–167: This is for enhanced warming over land for gridpoint perturbations if I am interpreting Fig. 4 left column correctly, right? Please indicate that you are referring to Fig. 4 in this statement for clarity.*

Thank you for pointing it out. We have mentioned it in the paper.

*Lines 174–177: I don't see much difference in Fig. S9 or Fig. S10- can you show a difference plot to highlight this more clearly?*

Sorry for the misunderstanding. When we refer to Figures S9 and S10, we mean to say that Figures S9 and S10 show that the 2005 October Flood (2055 October Flood) has a much stronger uplift (as shown by the green areas in the first row in Fig. S10) near the coastal region and much stronger onshore wind (as shown by the red areas in the first row in Fig. S9) compared with the 2006 May Flood (2056 May Flood) and the 2006 June Flood (2056 June Flood). In Figures S9 and S10, we can easily observe it by comparing the first row with the second and third rows.

*Lines 181–182: Again, I am failing to see where precipitable water is much higher in the regional mean simulation than the gridpoint. Can you provide a domain mean in Figure S2 and put those numbers in each panel?*

To better illustrate the precipitable water difference between PGW_T_regional and PGW_T_gp, we have plotted the figure below and added it to the supplement.

[Figure]

Period precipitable water (mm d$^{-1}$) and simulation difference

**Figure. R4 (Left) Period mean precipitable water (mm) over the whole domain from the simulation with temperature perturbation at regional mean scale. (Right) Difference between the period mean precipitable water (mm) from the gridpoint perturbation simulation and regional mean simulation from the left column.**

*Line 203–205: When you are saying that dynamical fields are assumed to be unchanged, I think you are speaking very specifically about uniform temperature perturbations in the regional sense. However, I do want to clarify that you have to be careful about making these statements since this is not the case in all PGW simulations. I think it is more accurate to say that the PGW method assumes that the dynamical changes are much smaller order magnitude than thermodynamic changes (Liu et al. 2016; O'Gorman 2015). Furthermore, it is impossible to assume the dynamics don't change due to change in thermodynamics (i.e., temperature), because these interact with each other. For example, Dougherty and Rasmussen (2021) show that storms with stronger updrafts show greater increases in rainfall in 4-km PGW simulations, likely due to the latent heat feedback suggested by Trenberth (1999).*

Sorry for the misunderstanding. By saying that dynamical fields are assumed to be unchanged, we mean that the dynamical field in the WRF input is not changed. Certainly, the modified thermodynamic fields will change the original dynamical fields as a result of mechanisms like the geostrophic adjustment. We have clarified this issue in the newly added discussion section and cited the reference you mentioned.

*Lines 209–210: How do you know PGW_T_regional is overestimating precipitation when there is no future "truth" or observations to compare it to? What is your baseline for this statement?*

While we cannot know for sure if PGW_T_regional overestimates the precipitation, we mean to say that PGW_T_regional overestimates precipitation compared with the gridpoint perturbation methods. However, there are good reasons to believe that PGW_T_regional is behaving unphysically. Because the PGW_T_gp employs the time-varying temperature perturbations at each gridpoint and pressure level (3D space + 1D time), it can capture the horizontal spatial variance of the temperature perturbation. But PGW_T_regional only uses the time-varying temperature perturbations at the regional mean scale at each pressure level (2D space + 1D time), and so does not include any information about the horizontal spatial variance of the temperature perturbation. There is ample evidence that the temperature perturbation over the land will be larger than the temperature perturbation over the sea, and so it is likely that the regional mean modification leads to future temperatures over the sea that are higher than what climate model projections indicate. The overestimated temperature perturbation in PGW_T_regional can be seen in Figure 4 (in the main paper). Since the relative humidity is fixed in our PGW runs, the change of specific humidity is determined by the temperature perturbation, and is subsequently larger in the regional mean simulation. Since precipitation in this region is largely driven by on-shore flow from the Atlantic, the enhanced specific humidity leads to enhanced vapor transport and increased moisture convergence (and subsequently higher precipitation in Figure 5 and 6 of

the paper). Therefore, we claim for this choice of domain, the simulated precipitation in PGW_T_gp is more reliable compared with PGW_T_regional.

*Line 226: Are you referring to the purple cross in this figure? If so, please state it as such.*

Yes, the enhanced precipitation refers to the purple cross in Figure 6 and we have stated it.

*Line 229: Is the reduction over the sea compared to the historical simulation or T-only experiment?*

It's compared with PGW_T_gp and we have clarified it in the paper.

*Lines 232–233: I think it's important to think about the larger view of why the wind magnitudes are stronger. The other two floods are warm-season events in May and June that have weaker synoptic forcing, whereas in October, synoptic dynamics and the forcing from it are quite strong. This would be my guess as to why the wind perturbation is stronger in the October flood than the others, and I think you should note that somewhere in your discussion*

We agree that understanding why the onshore wind magnitude increase is stronger in October is important; however, in this case it's mainly determined by the wind projection and the corresponding perturbation from CESM1 LE. Because this study focuses on how the PGW simulations respond to different perturbation modification methods, explaining what causes the wind projections in GCMs are out of the scope of this study. Other papers (for example, Kulkarni et al., 2014) also show that the surface wind speed is projected to increase more in the winter over NEUS in GCMs' projections. We have briefly discussed this in the discussion section.

*Line 235: Again, please be clear which marker you are referring to in Figure 6 when discussing these results.*

"Referring to the difference between the purple cross and green triangle in Fig. 6" has been added.

*Lines 262–263: That's not true- hypsometric equation states that a thicker layer = a warmer layer. However, by ideal gas law, $p = \rho RT$, pressure and temperature are proportional such that higher pressure = warmer temperature.*

As we stated, WRF uses a hybrid vertical coordinate which is of roughly constant layer thickness at ground level, and so near-surface changes to SLP are incorporated at constant volume. In the hypsometric equation ($Z_2 - Z_1 = \frac{R \times T_v}{g} \times \ln(\frac{P1}{P2})$), when $Z_2 - Z_1$ is fixed, an increasing $P_1$ indicates a decreasing $T_v$ (mean virtual temperature). The ideal gas law describes the state of a hypothetical ideal gas (air parcel), and it's not applicable here because, although density varies inversely with layer thickness, air density is not immediately known.

*Line 266–267: The colder temperatures and reduced precipitable water appears to be*

*true only over the sea surface.*

Note that precipitable water is defined as the depth of water in a column of the atmosphere instead of precipitable water in the surface atmosphere, and as we show in Figure 12 that there is a generally reduced precipitable water over the sea region instead of the surface of sea. In fact, most of the atmospheric moisture and precipitable water is present at the near surface. Therefore, a decrease in surface temperatures will cause the decrease in total precipitable water.

*Line 27–274: Were the wave-like noises found only in the paper you cite? Because I do not see any in Figure S6 or S7.*

Please refer to our response to the general issue 3. Figures S6 and S7 are the period mean plots and so do not exhibit the wave-like noise as this noise is not stationary; however, if we look at the animation of sea level pressure during flood periods (https://zenodo.org/record/6544880#.Y0YyQuzMJ9s), we can clearly observe the wave-like noise.

*Figure 14: Is this just the 2056 May mean temperature from LE CESM1 without running any PGW simulations?*

Yes, the black line illustrates the period mean temperature difference between the historical and future periods projected by the multi-ensemble mean of CESM LE.

*Line 290–292: I would add "stronger synoptically-forced systems" after frontal systems.*

This has been added.

**Technical comments:**

*Table 1: Please replace "huge" with "large", since the former is too colloquial.*

Corrections have been made.

*Figure 2:*
*The top left panel appears to have a different color scheme than the rest of the figure. Please fix this.*
*Caption: Is the average temperature the average monthly temperature? If so, please include that in the figure caption.*

Thank you for pointing it out. As suggested by reviewer 2, we plotted the temperature difference in the second and third columns in Figure 2. The average temperature is the period mean temperature during three flood periods (the periods we defined the Table 1). We have replaced the average temperature with period mean temperature in the caption.

*Lines 170–172: This is confusing as written. I would suggest rewriting to say "increase*

*from 12.69 mm/d to 13.86 mm/d in Oct 2005, 7.19 to 7.83 mm/d in May 2006, and 7.43*
*to 7.88 mm/d in June 2006)".*

We have modified the sentence as you suggested.

*Figure 4:*
*I would change the label of the left-column figures to PGW_T_gp -*
*PGW_T_regional since you are showing a difference.*
*Please flip the colorbar so it matches the intuition that red is warmer and blue is*
*cooler.*

In all period mean and simulation difference figures (like Figures 2, 3, 5, 8 and 9), the left
column shows the baseline simulation, while the rest of the columns indicate the differences
compared with the baseline (as stated in the captions). Otherwise, the subtitles might be too long.
For example, for the difference between *PGW_T_WIND_SLP_gp and PGW_T_gp, the subtitle*
*will be PGW_T_WIND_SLP_gp - PGW_T_gp which is lengthy and redundant.*

Thanks for your suggestion on color bar. We have reversed it.

*Figure 7 should be the last figure given that the first time it is mentioned is on Line 278.*

We have relocated it to the correct position.

*Figure 10: I think you mean hPa instead of "(mm/d)" in the figure caption, right?*

We have corrected the caption.

**REVIEWER COMMENTS - Referee Reviewer 2**

*This article deals with a hot topic that may suit HESS. This study examines the sensitivity and robustness of the PGW method over NEUS by conducting multiple PGW experiments. In addition, several PGW experiments are conducted to answer the three key questions related to the application of the PGW method. The results may help further understand the impact of different PGW simulations on climate projections. Overall, I think it is a pretty good job. However, some scientific or presentation issues need to be carefully addressed. Therefore, the reviewer recommends that this manuscript should be accepted after minor revision.*

Dear reviewer, thank you very much for your appreciation and suggestions. Certainly, these suggestions are very helpful in improving the scientific rigor and presentation of this work. We modified the paper correspondingly and please check the point-to-point responses below.

**Minor revision:**

*It is recommended to use consecutive line numbers.*

The line numbering is provided by the HESS LaTeX template. In the posted PDF line numbers appear to be consecutively numbered.

*It is suggested that some necessary statistical parameters should be provided to quantify the difference in precipitation and temperature simulation performance of different schemes.*

Thank you for your suggestions. We employed the K-S Test, which aims to determine if two data have the same distribution, and Student's t-test, which aims to examine if two data have the same mean value among all PGW simulations during the total simulation period. We chose the PGW_T_gp as the baseline and compared it with other PGW simulations on the regional mean precipitation and 2-meter temperature. Results show that the p-values from both the K-S Test and t-test are much larger than 0.05 in all cases, which indicates we cannot reject the null hypothesis that the regional mean precipitation and 2-meter temperature of PGW_T_gp and other PGW simulations have the same distribution and mean values at the 95% confidence level.

Further we conducted similar statistical tests on the precipitation and 2-meter temperature at each gridpoint. The results show that, except PGW_T_ZG_gp, the p-values of both K-S Test and t-test between all other PGW simulations and PGW_T_gp are nearly zero (much less than 0.05), indicating evidence against the null hypothesis that the precipitation and 2-meter temperature at each gridpoint of PGW_T_gp and other PGW simulations (except PGW_T_ZG_gp) have the same distribution and mean values at the 95% confidence level. However, the p-values of these two tests on precipitation and 2-meter temperature at each gridpoint between PGW_T_gp and PGW_T_ZG_gp are still much larger than 0.05 (0.9997 and 0.7403 for K-S Test and t-test), indicating that, even at gridpoint scale, PGW_T_ZG_gp's precipitation and 2-meter temperature still have the same distribution and mean values as PGW_T_gp.

This lends evidence to our claim that modifying ZG (geopotential height) in PGW simulations makes little difference on the final simulation and is not necessary. Also, the statistical tests further confirm our conclusion that different PGW methods will displace the weather events (the precipitation and 2-meter temperature at each gridpoint) but have a few impacts on the regional mean meteorological fields (the regional mean precipitation and 2-meter temperature).

 We have added the content above to the newly added discussion.

*The figures legend/caption is not self-explanation, which should be improved. In addition, many subgraphs in Figures 2 and 3 are very similar, making their differences challenging to identify. It is suggested to adopt the form of Figs.8-12 or add the statistical parameters mentioned in question 2.*

Thank you very much for pointing it out. We will improve the legends and captions to make them clearer. In Figures 2 and 3, we will plot the differences between our simulation and other datasets. Attached is the updated the Figure 3.

[Figure]

**Figure R1 Period mean precipitation (mm/d) and difference (the difference is calculated as the difference between each reanalysis/observational data and our simulation)**

*Figure 1 recommends that the macro location comes from the continent to geologically locate. The locations of the three regions should be marked in Figure 1*

Sorry, we're not sure if we understand this suggestion clearly. The three storms pass through our domain, along with their associated precipitation, as shown in the references in Table 1. In Figure 5 and Figure 8 (in the main paper), we can observe the distribution of precipitation during three storms and speculate about their centers; however, because they are noisy and spread out, it's difficult to directly specify the locations of these three storms and associated flooding on the domain figure.

*Please clarify the "returned" flood period (2055 April to 2056 July). Is this the 50-year return period for floods considered? Why not consider using other periods?*

The "returned" flood period (2055 April to 2056 July) in this paper means the scenario when the same larger-scale circulation and dynamic fields of the historical 2005 April to 2006 July flood period reoccurred under the RCP8.5 warming scenario in the middle of this century. The historical 2005 April to 2006 July flood period includes three major extreme precipitation / flood events over the NEUS. By simulating this consecutive period, we can reduce the spin-up period needed. We choose the middle of this century (2050s) as the simulated future period because it's widely used and studied.

*Please check whether Figure 13 is incorrect. Also, please adjust the color bars in Figures 11, 3, and S1, as some of them are not valid.*

Thanks a lot for pointing this out. Figure 13 should be correct and we changed its first column's color bar to improve its appearance; however, the caption of Figure 13 has one problem – it should be "over the inner domain" instead of "over the whole domain." We also adjusted the color bars in Figures 3, 11 and S1 as you suggest.

*The authors should rearrange the structures of the manuscript. The discussion is missing, maybe the result should change to result and discussion. You must buy the results from other similar studies in the discussions section. It is worth completing comparisons or differences with similar studies in other regions of the country or the world with related studies.*

Thank you for the suggestion. We will restructure the paper and add a discussion section. However, because our PGW method sensitivity study is novel, it is difficult to contextualize it in the literature. In fact, we haven't found any similar papers (we even consulted with Dr. Jimy Dudhia – one of the authors of WRF and he told us that this study is very novel). But we will try to elaborate the discussion with enough reference papers.

[Figure]

**Jimy Dudhia** <dudhia@ucar.edu>
to me, Paul

I have not seen someone study the difference in PGW adjustments by not varying both winds and thermodynamic fields. It would be interesting. For FDDA nudging people have studied nudging one or the other.
Jimy

*The authors should further clarify the shortcomings and limitations of the study.*
*Please check the format of the references to meet the journal's requirements.*

Agreed. We have identified the several shortcomings or limitations of the study and will discuss these in the text: for example, our simulation has a coarse resolution and uses cumulus scheme which reduce the model performance; the number of flood events studied is limited.

**Reference**

Schär, C., Frei, C., Lüthi, D., and Davies, H. C.: Surrogate climate-change scenarios for regional climate models, Geophysical Research Letters, 23, 669–672, 1996.

Frei, C., Schär, C., Lüthi, D., and Davies, H. C.: Heavy precipitation processes in a warmer climate, Geophysical Research Letters, 25, 1431–1434, 1998.

Karmalkar, A. V., Thibeault, J. M., Bryan, A. M., and Seth, A.: Identifying credible and diverse GCMs for regional climate change studies-case study: Northeastern United States, Climatic Change, 154, 367–386, 2019.

Sillmann, J., Kharin, V., Zhang, X., Zwiers, F., and Bronaugh, D.: Climate extremes indices in the CMIP5 multimodel ensemble: Part 1. Model evaluation in the present climate, Journal of Geophysical Research: Atmospheres, 118, 1716–1733, 2013.

Kay, J. E., Deser, C., Phillips, A., Mai, A., Hannay, C., Strand, G., Arblaster, J. M., Bates, S., Danabasoglu, G., Edwards, J., et al.: The Community Earth System Model (CESM) large ensemble project: A community resource for studying climate change in the presence of internal climate variability, Bulletin of the American Meteorological Society, 96, 1333–1349, 2015.

Ullrich, P., Xu, Z., Rhoades, A., Dettinger, M., Mount, J., Jones, A., & Vahmani, P. (2018). California's drought of the future: A midcentury recreation of the exceptional conditions of 2012–2017. Earth's Future, 6 (11), 1568–1587.

Jin, J., Miller, N. L., & Schlegel, N. (2010). Sensitivity study of four land surface schemes in the WRF model. Advances in Meteorology, 2010.

Case, J. L., Crosson, W. L., Kumar, S. V., Lapenta, W. M., & Peters-Lidard, C. D. (2008). Impacts of high-resolution land surface initialization on regional sensible weather forecasts from the WRF model. Journal of Hydrometeorology, 9 (6), 1249–1266.

Sujay Kulkarni, Huei-Ping Huang, "Changes in Surface Wind Speed over North America from CMIP5 Model Projections and Implications for Wind Energy", Advances in Meteorology, vol. 2014, Article ID 292768, 10 pages, 2014. https://doi.org/10.1155/2014/292768

---

## Author Response (AR2)

**Response letter to Referee Reviewer 1**

**REVIEWER COMMENTS**

Referee Reviewer 1

*I appreciate the time the authors took to address my concerns and increase the robustness of their findings. Overall, I find that the manuscript has been greatly improved and thus I have mostly minor comments to suggest.*

Dear reviewer, thank you very much for your suggestions which were very helpful in improving the presentation of this work. For the gravity wave section, we have opted to keep most of the content, as it is important to understand gravity waves in PGW runs. Our use of hourly animations and magnitude spectrum can provide some clarity on this issue since previous studies only used mean fields to infer the absence of gravity waves. However, to address your concern, we have rewritten the paragraph to clarify our explanations and interpretations of the results. We have also modified the paper following your comments in our point-by-point responses below.

**General:**

• *I am confused about the point of Section 3.3 "Gravity wave noise" because the authors contradict the claims they make in this section with other parts of the paper. In this section, they attribute the appearance of gravity waves to storm development, which I believe is accurate and not noise especially because in the introduction they state that spurious gravity waves could be generated from the boundary of the domain if the perturbation upsets geostrophic balance (lines 61–63). Clearly, these are not just "spurious" gravity waves, but rather real artifacts due to storm intensification upon watching the animations sent by the authors. In the conclusions (lines 356–358), they contradict Section 3.3 and state that all experiments produce spurious waves. You cannot state that these are both spurious and storm-generated gravity waves, they must be one or the other (unless there are other gravity waves I did not see).*

*From this point, I think there is too much speculation around the appearance of waves and I would recommend simply stating that they exist, and could be due to stronger storms in the PGW simulations due to the warmer temperatures. Or you could just state that they exist and are stronger with some perturbed variables but not others. If these figures are just averaged over the time period of the flood (including when it is raining), again, you are introducing mesoscale modifications of the perturbations, including mesolows, outflows, and cool pools. It is impossible to evaluate these figures and what the changes in pressure, temperature, and CAPE mean if they are already being contaminated by the presence of storms.*

Sorry for the confusion and thank you for urging us to clarify our key points on gravity waves in the simulations. Here we would like to further clarify the gravity waves discussed in subsection 3.3. First, as stated in the manuscript and our previous response letter, the storm development contributes to and magnifies the gravity waves; however, storm development is not necessarily

the dominant or the only deterministic source of the obvious gravity waves in the PGW simulations. As shown in Figures R1 and R2, the magnitude and frequency of gravity waves are negligible or much lower in the historical run compared with the PGW runs. The more obvious gravity waves in the PGW runs may be due to the development of a stronger storm under warming, geostrophic imbalance caused by the perturbed boundary conditions, and/or the interactions between the two factors. If the most significant gravity waves are solely caused by stronger storms in the PGW simulations and are due to the warmer temperatures, then the magnitude increase of gravity waves should be comparable to the storm intensification (e.g. the increased precipitation and associated increased latent heating). However, the magnitude and frequency of gravity waves in the historical run are much lower compared with those in the PGW runs. That is, the gravity waves are amplified far more than can be explained by the intensification of the storms.

Additionally, we can see from Figure R3 that the gravity waves during the 2006/2056 March dry periods are not obvious in both the historical/PGW runs, which suggests that storms can magnify the gravity waves in the presence of geostrophic imbalance in the boundary conditions. In other words, the gravity waves in the PGW runs are contributed by the inconsistencies and imbalances between the outer and inner domains due to the PGW perturbations and magnified by the occurrence of storms. This behavior not only increases the magnitude of gravity waves but also increases the imbalance between the boundary conditions and the inner domain. With that said, we admit that the analysis of gravity waves is still not comprehensive so further studies are needed to deepen our understanding.

We recognize that the paragraphs in the manuscript do not convey this clearly. We have modified them, instead of shortening them, as we believe that the discussion of gravity waves in PGW experiments is important for the regional modeling community. Specifically, many PGW studies preferred to add a regional mean warming signal to reduce the gravity waves caused by unrealistic adjustments to geostrophic imbalances (Hill and Lackmann, 2011; Yates et al., 2014; Mallard et al., 2013b; Ullrich et al., 2018). Their assumption is that, by modifying the temperature at each grid point according to the GCMs' projections instead of using regional mean perturbations, the temperature gradient and geostrophic balance in the control run based on reanalysis boundary conditions will no longer hold, so gravity waves will be excited due to adjustments needed to restore the geostrophic balances. But here we can see that in both PGW_T_regional and PGW_T_gp, the magnitudes of the gravity waves are comparable and are both much larger than the waves in the historical run, demonstrating that PGW_T_regional does not obviously reduce the gravity waves as suggested. Also, several previous papers (e.g. Lackmann (2013) and Mahoney et al. (2018)) concluded that gravity waves were not found in their PGW runs, even though they only analyzed the period mean fields, which mask the appearance of gravity wave noise, and do not provide animations or magnitude spectrum as provided in this study. The appearance of gravity waves is a crucial question in PGW studies so we hope our work can shed some light on this issue and call for more attention to this phenomenon in future work.

The corresponding paragraph is modified as follows:

In WRF, boundary conditions are specified based on the input forcing (here the historical reanalysis data with climate perturbations) and the numerical solutions are nudged towards the imposed boundary conditions within the buffer zone of the outer-most domain (Skamarock et al., 2008). Within the inner domain, except for the most peripheral grid points, meteorological fields are much less constrained by the boundary conditions and can significantly depart from the forcing data (Skamarock et al., 2008). This can induce inconsistencies among the meteorological fields in the outer and inner domains and explain why the wave-like noise also appears in PGW_T_regional (refer to the animations at Xue and Ullrich (2022)), even though the geostrophic balance holds in the initial and boundary conditions. From the animations of SLP during the October flood event (Xue and Ullrich, 2022), we can infer that the gravity wave is not solely excited by the storms, as the magnitude of the gravity wave is negligible in the historical run compared to the PGW run even though the storm is also present in the historical run. Furthermore, gravity waves are amplified in the PGW run more than expected from the difference in the precipitation between the PGW and historical runs. Similar results can be observed in the magnitude spectrum of SLP (Fig. S15). Storms also play an essential role in magnifying the gravity waves through significant advection of energy and momentum, as it is apparent that gravity waves are much stronger during storms than during periods with nearly no precipitation (refer to the animations at Xue and Ullrich (2022)). We conclude that the amplified gravity waves in the PGW run during storm events reflect the interactive effect of inconsistency between the meteorological fields in the outer and inner domains and the excitation of gravity waves by the storms. Although gravity waves in the PGW simulations are inspected through hourly animations and magnitude spectrum, the analysis of gravity waves could be the basis of its own standalone study.

Please refer to https://zenodo.org/record/6544880#.Y0YyQuzMJ9s
**Figure. R1 The animation of sea level pressure during the 2005 October Flood and returned 2055 October Flood**

**The Sea Level Pressure Field and Its Magnitude Spectrum after Fourier Transform**

Sea Level Pressure in Historical Run on 2005 Oct 12th 00:00

[Figure]

Magnitude Spectrum in historical Run on 2005 Oct 12th 00:00

[Figure]

Sea Level Pressure in PGW_T_gp Run on 2005 Oct 12th 00:00

[Figure]

Magnitude Spectrum in PGW_T_gp Run on 2055 Oct 12th 00:00

[Figure]

Sea Level Pressure in PGW_T_WIND_gp Run on 2005 Oct 12th 00:00

[Figure]

Magnitude Spectrum in PGW_T_WIND_gp Run on 2055 Oct 12th 00:00

[Figure]

**Figure. R2 The sea level pressure field and its magnitude spectrum after Fourier transform**

Please refer to https://zenodo.org/record/6544880#.Y0YyQuzMJ9s
**Figure. R3 The animation of sea level pressure during the 2006 March dry period and returned 2056 March dry period**

**Specific comments**

*• Lines 12–13: I do not think you can state the preservation of geostrophic balance does not hold, given my comment about gravity waves.*

Please refer to our response to the general comment. It's apparent in Figures R1 and R2 that the gravity waves are much stronger in the PGW runs compared with the historical run and the difference is much larger than expected from the difference in precipitation between the two runs, suggesting that both inconsistencies and imbalances between the inner and outer domain brought by the PGW perturbations and the intensified storm development in the PGW runs contribute to the amplified gravity waves in the PGW runs. We have modified the paragraph to provide clearer explanations and mentioned that our analysis is still not comprehensive and further analysis is needed.

*• Line 48, starting with "In these first PGW", I recommend starting a new paragraph since the one you have is already quite long.*

Thanks a lot for your suggestions. We started a new paragraph as suggested.

*• Line 136: How are perturbations linearly interpolated? Do you mean the perturbations between months?*

Yes, we have added "(between the middle of two consecutive months)" after "and linearly interpolated in time" to clarify it.

*• Line 138–140: Are you taking the CESM 40-member ensemble mean for your future forcing? If so, that should be stated here.*

Sorry for the misunderstanding, and we did state that the CESM ensemble mean is used to provide perturbations in the subsection – Methodology and modified forcings – "For our PGW simulations of the future, the initial and boundary conditions are adjusted by adding long-term monthly mean of the ensemble mean climate perturbations from the Community Earth System Model (CESM1) Large Ensemble (LE) dataset" (Line 132).

*• Figure 2 caption: I don't see a column with the difference with IMERG. Please remove this from the caption or include figures showing this difference.*

Thanks for pointing out the typos. We have deleted the mention of IMERG from the caption.

*• Line 189–191: I understand how regional vs gridpoint perturbations in T would cause an impact in the October case due to the reduced land-sea contrast in the regional perturbation, but I don't understand why regional vs. grid point perturbations would affect the frontal system? Do you think it's due to a reduced horizontal temperature gradient using the regional perturbations? If so, these sorts of interpretations would be helpful to discuss in order to make sense of your results.*

We have added "Additionally, as the frontal system's intensity is highly dependent on the horizontal temperature gradient (Sawyer, 1956; Bosart, 1975; Reeder et al., 2021), it's intuitive that the first flood event is sensitive to the different spatial scales of temperature perturbations applied to the boundary conditions." But to confirm whether using regional perturbations reduces the horizontal temperature gradient and hence the strength of frontal systems requires additional analysis which is out of the scope of this study."

Instead, In Lines 189-191, we attributed the larger precipitation increase in the first event to two factors. We know that the scaling of extreme precipitation with temperature can be decomposed into two factors: vertical pressure velocity and the vertical derivative of the saturation specific humidity (Pfahl et al., 2017). As the first event has a more intense uplift (Fig. S10), it will have a larger response to the increasing humidity under warming. Also, this event has a stronger on-shore flow (Fig. S9) which means it can transport more precipitable water from the ocean to the coastal area where precipitation mainly occurs.

*• Figure 3: Please state that all precipitation products are regridded to the same grid-spacing and what that grid-spacing is.*

Thanks for the advice. We have added "All datasets have been interpolated to the resolution of our inner domain (10km)." to its caption.

*• Figure 4: Please add "difference" after "PGW_T_gp" in the figure headings in the right column to make it clear that this is a difference plot.*

Since our current title and captions already point out which columns are different plots and we follow the same layout in all similar plots (e.g. Figures 7 - 12), we'd prefer not to add "difference" to the figure headings to prevent them to be too long (e.g. PGW_T_WIND_ZG_SLP_gp difference).

*• Figure 6: The titles overlap between the two columns, so I recommend using a smaller font to make sure they do not overlap.*

We have adjusted the plot following your advice.

*• Lines 240–241, 246–247: Are you referring to Fig. S6 and S7 here? If so, please add those citations.*

Thanks for pointing out the need to add citations. We have added the corresponding citations and clarifications.

*• Lines 283–284/ "Note that precipitable water is defined as the depth of water in a column of the atmosphere instead of precipitable water in the surface atmosphere": I am well aware of how precipitable water is defined. I was simply pointing out that the reduction in precipitable water for the PGW_T_SLP_gp experiment mostly appears over the sea rather than the land (except for the 2056 May case), which is worth stating in your results.*

Sorry for the misunderstanding. That's an interesting observation which may be because in the second event SLP increases less over the sea (Figure 9), which further illustrates the impact of SLP perturbations on surface temperature. We have added "The reduction in precipitable water over the sea is stronger than over the land, except for the 2056 May flood period, probably because the SLP increase over the sea is smaller during this event (Fig. 9)."

*• Line 292: I recommend reordering the figures so that Fig. 9 comes after Fig. 13 since this is the first time you mention it.*

Thank you for your suggestions. We have reordered these two figures.

*• Lines 297–300: It is well known that CESM-LE tends to have a very strong warming signal in RCP8.5 compared to other GCMs, so what if the inclusion of SLP perturbations helps to ameliorate those biases?*

CESM-LE projects a stronger warming signal under RCP8.5 compared to other GCMs, but the stronger warming is not a model bias that should be corrected because climate sensitivity is an emergent behavior of climate models. CESM1 has been demonstrated to be a high-quality model and using CESM-LE helps reduces uncertainties due to internal variability (Kay et al., 2015; Swann et al., 2016; Sillmann et al., 2013; Karmalkar et al., 2019). Therefore CESM-LE is widely used in climate studies (Labe et al., 2018; Zheng et al., 2018; Bellomo et al., 2018; Peings et al., 2017).

To reduce model uncertainties in projections due to model biases, the multi-model mean may be used to produce the climate perturbations as adopted in some previous studies instead of including SLP perturbations to counter the model biases. As we stated in the manuscript, the warming induced by the SLP perturbations is more likely due to unrealistic adjustments made within WRF according to the hypsometric equation because we can clearly see from Figure 13 and 14 that the warming signals in PGW_T_SLP_gp deviate from the projections of CESM-LE and are inconsistent at the surface level.

*• Supplementary Table S1: you are missing a citation after CLM4*

We have added the corresponding citation.

**Reference**

Labe, Z., Magnusdottir, G., & Stern, H. (2018). Variability of Arctic sea ice thickness using PIOMAS and the CESM large ensemble. Journal of Climate, 31(8), 3233-3247.

Zheng, X. T., Hui, C., & Yeh, S. W. (2018). Response of ENSO amplitude to global warming in CESM large ensemble: uncertainty due to internal variability. Climate Dynamics, 50, 4019-4035.

Bellomo, K., Murphy, L. N., Cane, M. A., Clement, A. C., & Polvani, L. M. (2018). Historical forcings as main drivers of the Atlantic multidecadal variability in the CESM large ensemble. Climate Dynamics, 50, 3687-3698.

Peings, Y., Cattiaux, J., Vavrus, S., & Magnusdottir, G. (2017). Late twenty-first-century changes in the midlatitude atmospheric circulation in the CESM large ensemble. Journal of Climate, 30(15), 5943-5960.

Mahoney, K., D. Swales, M. J. Mueller, M. Alexander, M. Hughes, and K. Malloy. (2018). An examination of inland-penetrating atmospheric river flood event under potential future thermo- dynamic conditions. J. Climate, 31, 6281–6297, https://doi.org/ 10.1175/JCLI-D-18-0118.1.

Lackmann, G. M.. (2013). The south-central U.S. flood of May 2010: Present and future. J. Climate, 26, 4688–4709, https://doi.org/10.1175/JCLI-D-12-00392.1.

Schär, C., Frei, C., Lüthi, D., and Davies, H. C.: Surrogate climate-change scenarios for regional climate models, Geophysical Research Letters, 23, 669–672, 1996.

Frei, C., Schär, C., Lüthi, D., and Davies, H. C.: Heavy precipitation processes in a warmer climate, Geophysical Research Letters, 25, 1431–1434, 1998.

Kay, J. E., Deser, C., Phillips, A., Mai, A., Hannay, C., Strand, G., Arblaster, J. M., Bates, S., Danabasoglu, G., Edwards, J., et al.: The Community Earth System Model (CESM) large ensemble project: A community resource for studying climate change in the presence of internal climate variability, Bulletin of the American Meteorological Society, 96, 1333–1349, 2015.

Ullrich, P., Xu, Z., Rhoades, A., Dettinger, M., Mount, J., Jones, A., & Vahmani, P. (2018). California's drought of the future: A midcentury recreation of the exceptional conditions of 2012–2017. Earth's Future, 6 (11), 1568–1587.

Mallard, M. S., Lackmann, G. M., and Aiyyer, A.: Atlantic hurricanes and climate change. Part II: Role of thermodynamic changes in decreased hurricane frequency, Journal of climate, 26, 8513–8528, 2013a.

Hill, K. A. and Lackmann, G. M.: The impact of future climate change on TC intensity and structure: A downscaling approach, Journal of Climate, 24, 4644–4661, 2011.

Yates, D., Luna, B. Q., Rasmussen, R., Bratcher, D., Garre, L., Chen, F., Tewari, M., and Friis-Hansen, P.: Stormy weather: Assessing climate change hazards to electric power infrastructure: A Sandy case study, IEEE Power and Energy Magazine, 12, 66–75, 2014.

Pfahl, S., O'Gorman, P. A., and Fischer, E. M.: Understanding the regional pattern of projected future changes in extreme precipitation, Nature Climate Change, 7, 423–427, 2017.

Xue, Z. and Ullrich, P.: The sea level pressure animations during flood and dry periods, 10.5281/zenodo.6544880, 2022.